# Learning Reward–Cost Balance in Safe RL via Score-Based World Models

Yuetian Wang [1]  Dianxi Shi [2]  Yuanze Wang [1]  Huanhuan Yang [3]  Shiming Song [2]  Chunping Qiu [2]

## Abstract

Safe reinforcement learning (Safe RL) seeks to optimize long-term performance while ensuring adherence to safety constraints. However, most existing approaches address safety in a simplified manner, typically by linearly combining rewards and costs, which provides limited guidance when safety and performance interact in complex, nonlinear ways. We present USB-RL (Unsupervised Score-Balanced Reinforcement Learning), a model-based framework that learns implicit safety–performance preferences directly from experience. Our approach infers a monotone partial-order score through self-supervised pairwise comparisons of long-horizon outcomes—requiring no human preference labels—capturing nuanced trade-offs without relying on manually tuned cost weights. The learned score guides model-based policy optimization by dynamically balancing safety and performance, enabling flexible and adaptive multi-step planning in imagination-based control. Across diverse safety benchmarks, USB-RL achieves strong returns while substantially reducing safety violations, demonstrating stable and interpretable safety–performance trade-offs.

## 1. Introduction

Safe reinforcement learning (Safe RL) aims to optimize long-term reward while satisfying safety constraints, typically formulated as a Constrained Markov Decision Process (CMDP). Unlike model-free safe RL methods that rely on direct environment interaction and cost penalties (García & Fernández, 2015; Ray et al., 2019), model-based approaches learn a predictive world model, enabling counterfactual reasoning about future trajectories. This allows agents to simulate and avoid unsafe outcomes during planning, leading to

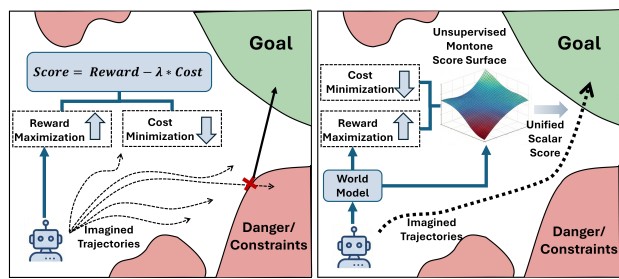

Figure 1. Architecture overview. USB-RL is a safe RL framework that integrates a learned, structured safety–performance score into model-based visual RL for continuous control.

more stable and sample-efficient trade-offs between reward and safety. However, existing model-based RL methods do not adequately address the key challenge in CMDP – the **reward-cost balance** (Roijers et al., 2013; Hayes et al., 2022; Huang et al., 2024). Although predictive rollouts can anticipate risky outcomes, safety is typically handled in a simplistic manner: Most approaches linearly penalize cost against reward, offering limited guidance when balancing nuanced trade-offs between safety and performance (Achiam et al., 2017; Dalal et al., 2018; Ray et al., 2019; Huang et al., 2024).

We argue that safe model-based RL requires a **unified and structured notion of risk preference** over imagined trajectories. Therefore, we introduce **USB-RL (Unsupervised Score-Balanced RL)**, a model-based framework that learns a unified, structured safety–performance preference end-to-end without human preference labels, and directly embeds it into the world model for imagination-driven planning. Here, "unsupervised" refers to the absence of human preference annotations; the score-learning signal is constructed self-supervisedly from reward-cost outcomes. The main methodological contributions of USB-RL are summarized as follows:

- *Self-supervised monotone preference learning.* We learn a structurally monotone reward–cost score from long-horizon returns via self-supervised pairwise dominance/tie constraints, requiring no human preference labels and yielding a stable ordering signal beyond linear penalties.

- *Score distillation for imagination-time control.* We distill the learned score into the world model with a distributional score head trained from an EMA teacher

[1]Shanghai Jiao Tong University, Shanghai, China [2]Intelligent Game and Decision Lab, Beijing, China [3]Beijing Academy of Science and Technology, Beijing, China. Correspondence to: Dianxi Shi <dxshi@nudt.edu.cn>.

*Proceedings of the 43rd International Conference on Machine Learning*, Seoul, South Korea. PMLR 306, 2026. Copyright 2026 by the author(s).

on stop-gradient latents, enabling efficient scoring in imagined rollouts.

- *ScoreCritic for long-horizon planning.* We learn a ScoreCritic to temporalize per-step scores and use it for score-ranked planning under strict cost-feasibility gating.

We empirically validate USB-RL on vision-based Safety-Gymnasium tasks and the MetaDrive driving benchmark. The results indicate that USB-RL achieves high returns while substantially reducing safety violations, demonstrating stable and interpretable safety–performance trade-offs.

## 2. Preliminaries

### 2.1. Constrained Markov Decision Process

Safe reinforcement learning is typically formalized as a constrained Markov decision process (CMDP) $\mathcal{M} = (\mathcal{S}, \mathcal{A}, P, R, C, \mu, \gamma, b)$ (Altman, 1995). Here $\mathcal{S}$ and $\mathcal{A}$ denote the state and action spaces, respectively. The transition kernel is $P(s' \mid s, a)$. At each step, the agent receives a scalar reward $r_t = R(s_t, a_t)$ and a scalar cost $c_t = C(s_t, a_t)$. A stationary parameterized policy $\pi_\theta(a \mid s)$ induces the discounted reward return and discounted cost return:

$$
\begin{aligned}
J^R(\pi_\theta) &= \mathbb{E}_{s_0 \sim \mu,\, a_t \sim \pi_\theta,\, s_{t+1} \sim P} \left[ \sum_{t=0}^{\infty} \gamma^t r_t \right], \\
J^C(\pi_\theta) &= \mathbb{E}_{s_0 \sim \mu,\, a_t \sim \pi_\theta,\, s_{t+1} \sim P} \left[ \sum_{t=0}^{\infty} \gamma^t c_t \right].
\end{aligned}
\tag{1}
$$

The CMDP objective is to maximize return while satisfying an expected discounted cost budget:

$$
\begin{aligned}
\pi^\star &= \arg \max_{\pi_\theta \in \Pi^C} J^R(\pi_\theta), \\
\Pi^C &\triangleq \left\{ \pi_\theta \in \Pi \mid J^C(\pi_\theta) \leq b \right\}.
\end{aligned}
\tag{2}
$$

We focus on the common single-constraint setting used by our benchmarks; extending the notation to multiple expected cost constraints is straightforward but omitted for clarity.

### 2.2. Model-Based Safe Reinforcement Learning

Model-based safe RL learns a parameterized world model $P_\phi(\cdot \mid s, a)$ from data and optimizes policies using imagined rollouts under $P_\phi$. Given $P_\phi$, the model-based reward and cost returns are defined analogously:

$$
\max_{\pi_\theta \in \Pi} J_\phi^R(\pi_\theta) \quad \text{s.t.} \quad J_\phi^C(\pi_\theta) \leq b,
\tag{3}
$$

where expectations in $J_\phi^R$ and $J_\phi^C$ are taken over trajectories with $s_0 \sim \mu$, $a_t \sim \pi_\theta(\cdot \mid s_t)$, and $s_{t+1} \sim P_\phi(\cdot \mid s_t, a_t)$. Throughout the paper, we use $\theta$ for policy parameters and $\phi$ for world-model parameters; we reserve $\omega$ for the score-function parameters introduced in Sec. 3.2 to avoid overloaded notation.

## 3. Method

### 3.1. Overview

USB-RL builds on a DreamerV3-style world model and learns a *unified, structured* safety–performance preference that is available during imagination. At a high level, we first learn a monotone ScoreNet from self-supervised pairwise ordering induced by long-horizon reward/cost returns, then distill this preference into the world model as a native score head so it can be queried efficiently inside imagined rollouts. Finally, we temporalize per-step scores with a ScoreCritic and use them to shape the actor update, while the planner retains strict cost-feasibility gating and uses the score only to rank feasible candidates. Training interleaves four coupled components:

- **World model learning** (Sec. 3.2): DreamerV3 world model with standard reconstruction/dynamics regularization and reward/cost prediction.

- **Unsupervised score learning** (Sec. 3.3): monotone ScoreNet trained by pairwise ordering from long-horizon reward/cost returns using real and imagined summaries.

- **Score distillation** (Sec. 3.4): transfer ScoreNet outputs into the world model as a distributional score head on stop-gradient latents for efficient imagination-time scoring.

- **Policy learning and planning** (Sec. 3.5): ScoreCritic temporalizes per-step scores for actor shaping; planning retains OSRP-style strict safety gating and ranks only cost-feasible rollouts by score return.

### 3.2. Model Components

USB-RL builds on a SafeDreamer-style Dreamer world model with imagination-based actor–critic learning (Fig. 2). The world model is an Recurrent State-Space Model (RSSM) (Hafner et al., 2023) with a deterministic hidden state $h_t$ and a discrete stochastic latent $z_t$; we denote the latent state by $s_t \triangleq (h_t, z_t)$.

**World model (RSSM).**

$$
\begin{cases}
\text{Observation encoder:} & z_t \sim E_\phi(z_t \mid h_t, o_t) \\
\text{Sequence model:} & h_t, \hat{z}_t \sim S_\phi(h_{t-1}, z_{t-1}, a_{t-1}) \\
\text{Observation decoder:} & \hat{o}_t \sim O_\phi(\cdot \mid s_t) \\
\text{Reward head:} & \hat{r}_t \sim R_\phi(\cdot \mid s_t) \\
\text{Cost head:} & \hat{c}_t \sim C_\phi(\cdot \mid s_t) \\
\text{Score head (new):} & \hat{y}_t \sim Y_\xi(\cdot \mid \tilde{s}_t)
\end{cases}
\tag{4}
$$

where $\tilde{s}_t = \mathrm{sg}(s_t)$ is the stop-gradient latent used to isolate dynamics learning (Sec. 3.3).

**Actor–critic agent.**

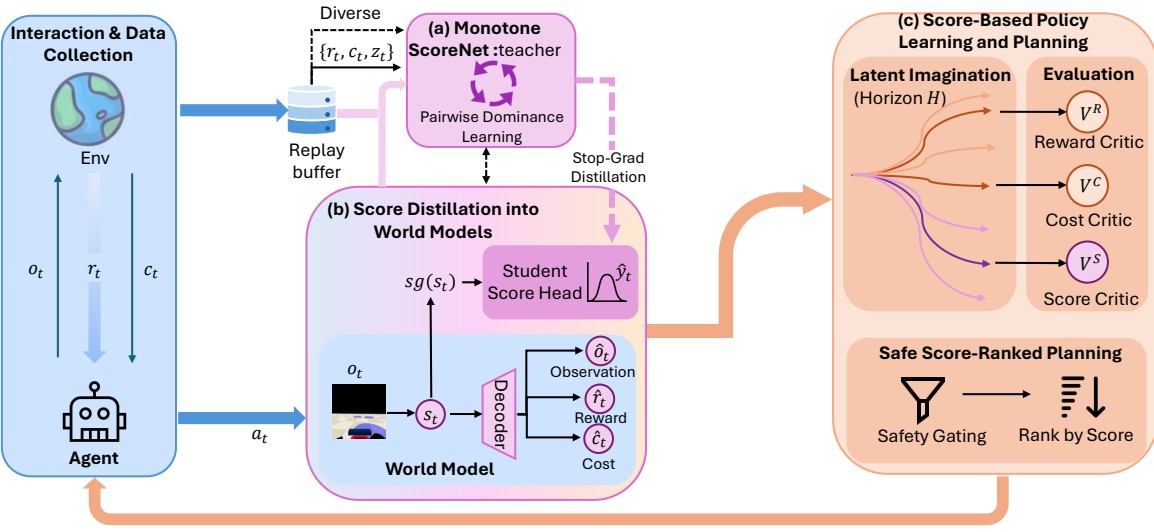

Figure 2. *Overview of USB-RL.* The agent collects transitions into a replay buffer; (a) a monotone ScoreNet learns a single-step safety–performance preference from pairwise dominance over long-horizon outcomes; (b) the learned preference is distilled into the world model as a stop-gradient student score head; (c) during imagination, reward/cost/score critics evaluate rollouts and the planner enforces cost-feasibility gating and ranks feasible candidates by score to select the action.

$$
\begin{cases}
\text{Actor:} & a_t \sim \pi_\theta(\cdot \mid s_t) \\
\text{Reward critic:} & \hat{v}_t^R \sim V_{\psi_r}(\cdot \mid s_t) \\
\text{Cost critic:} & \hat{v}_t^C \sim V_{\psi_c}(\cdot \mid s_t) \\
\text{ScoreCritic (new):} & \hat{v}_t^S \sim V_{\psi_s}^S(\cdot \mid s_t)
\end{cases}
\tag{5}
$$

**External ScoreNet (teacher).**

$$
\text{ScoreNet (new):} \quad y_t^\star = f_\omega(r_t, c_t, z_t) \tag{6}
$$

ScoreNet is trained with self-supervised pairwise ordering (Sec. 3.3), and its outputs $y_t^\star$ are distilled into the world-model score head (Sec. 3.4).

ScoreNet $f_\omega$ serves as an *external teacher* that learns a single-step safety–performance preference under rich conditioning $(r_t, c_t, z_t)$ using self-supervised pairwise ordering. To make this preference available efficiently during imagination—where $(r_t, c_t)$ are not directly observed and rollouts require many score queries—we *distill* ScoreNet outputs into a native world-model score head $Y_\xi$ on stop-gradient latents. Finally, the ScoreCritic $V_{\psi_s}^S$ *temporalizes* these per-step scores into a long-horizon value/return signal used for score-ranked planning and actor shaping.

### 3.3. Monotone ScoreNet

We learn a single-step safety–performance preference score with a parameterized function $f_\omega(r, c, z) : \mathbb{R} \times \mathbb{R} \times \mathbb{R}^d \to \mathbb{R}$, where $r$ is the immediate reward, $c$ is the immediate cost, and $z$ is a latent feature from the world model (the stochastic component of $s_t$). Although the score is evaluated on local

tuples $(r, c, z)$, it is supervised by long-horizon reward–cost outcomes, so that it captures how local reward, cost, and latent context relate to future safety-performance trade-offs. The score is designed to be *structurally monotone* with respect to the explicit reward and cost inputs:

$$
\frac{\partial f_\omega}{\partial r}(r, c, z) \geq 0, \qquad \frac{\partial f_\omega}{\partial c}(r, c, z) \leq 0, \ \ \forall (r, c, z). \tag{7}
$$

**Architecture.** To satisfy Eq. (7) while retaining state-dependent expressivity, we implement $f_\omega$ with three branches: (i) a reward branch $g_r(r)$ that is monotone non-decreasing in $r$, (ii) a cost branch $g_c(c)$ that is monotone non-decreasing in $c$ but enters the score with a negative sign, and (iii) an unconstrained context branch $h(z)$ that captures latent state information not represented by the immediate reward and cost. The reward and cost branches are fused with non-negative coefficients, so that global monotonicity with respect to $r$ and $c$ is preserved:

$$
f_\omega(r, c, z) = \tanh\Big(\alpha_r^\top g_r(r) - \alpha_c^\top g_c(c) + \beta^\top h(z) + b\Big),
\tag{8}
$$

where $\alpha_r, \alpha_c \succeq 0$. In implementation, $g_r$ and $g_c$ are univariate monotone networks built from non-negative-weight dense layers and monotone activations, with non-negative weights parameterized by a softplus transform; $\alpha_r$ and $\alpha_c$ are constrained similarly. Since $\tanh(\cdot)$ is monotone increasing, Eq. (8) ensures that, for any fixed $z$, the score

is non-decreasing in $r$ and non-increasing in $c$. The context branch $h(z)$ is unconstrained because monotonicity is defined with respect to the explicit reward and cost inputs while holding $z$ fixed; it can shift state-dependent score baselines but cannot reverse the reward/cost directions enforced by Eq. (7).

**Why a monotone score?** The monotone score is *not* intended as another soft reward–cost penalty. Strict safety is primarily enforced by cost-feasibility gating during planning, and the score is used only to rank candidates that already satisfy, or nearly satisfy, the safety budget (Sec. 3.5). Compared with fixed linear penalties or Lagrangian coefficients, which impose a single global reward–cost exchange rate, the learned score provides a state-dependent ordering signal while preserving the safe-RL semantics that higher reward should not reduce preference and higher cost should not increase preference. Architectural monotonicity is also complementary to the pairwise loss: dominance and tie losses constrain only sampled pairs, whereas the monotone parameterization acts as a global inductive bias for nearby regions encountered during imagination.

**Self-supervised pairwise supervision from long-horizon outcomes.** Training does not require human preference labels: supervision is self-generated from the agent's long-horizon reward and cost return estimates. Each training element is a summary $(x_i, G_i^r, G_i^c)$, where $x_i = (r_i, c_i, z_i)$ and $(G_i^r, G_i^c)$ are TD($\lambda$) returns for reward and cost, computed using the current reward/cost critics (with EMA targets for stability). We construct elements from two sources: (i) *real* summaries sampled from the replay buffer; and (ii) *imagined* summaries generated from a fixed reference latent state (the root of an imagination tree). Concretely, following DreamerV3, the observation encoder produces a posterior over a discrete stochastic latent, while the recurrent sequence model maintains a deterministic hidden state and predicts a prior for the next latent. For each root latent, we draw $K_{\text{roll}}$ stochastic continuations and compute TD($\lambda$) returns for each. During training, we primarily form *within-root* comparisons among continuations that share the same root, so that pairwise relations reflect policy/dynamics variability rather than state mismatch. We refer to this mechanism as *within-root pair construction*: pairs are formed mainly among continuations sharing the same imagination root, rather than sampled uniformly at random from the global pool of summaries. This is the structured pair construction whose effect we isolate in the ablation study (Sec. 4.5).

**Partial-order construction.** Given two elements $(i, j)$ with returns $(G_i^r, G_i^c)$ and $(G_j^r, G_j^c)$, define $\Delta G_{ij}^r = G_i^r - G_j^r$ and $\Delta G_{ij}^c = G_i^c - G_j^c$. We declare that $i$ *dominates* $j$ if $\Delta G_{ij}^r \geq \epsilon_r$ and $\Delta G_{ij}^c \leq -\epsilon_c$. We treat $(i, j)$ as *incomparable* if (a) reward and cost move in the same direction, i.e., $\Delta G_{ij}^r \Delta G_{ij}^c > 0$, or (b) both differences are within small tolerances, i.e., $|\Delta G_{ij}^r| < \epsilon_r$ and $|\Delta G_{ij}^c| < \epsilon_c$. The thresholds $(\epsilon_r, \epsilon_c)$ suppress noisy or practically insignificant comparisons; we anneal them from a coarse value to a smaller value during training as the value estimates sharpen (see Appendix for details).

**Losses.** Let $u_i = f_\omega(x_i)$. A dominance hinge loss enforces strict ordering with a margin $m$:

$$\mathcal{L}_{\text{dom}} = \mathbb{E}_{(i,j) \sim \mathcal{D}} \Big[ \max\{0,\, m - (u_i - u_j)\} \Big], \quad (9)$$
$$x_k = (r_k, c_k, z_k).$$

A tie loss discourages spurious preferences on incomparable pairs by compressing their score gap into a narrow band of width $\delta$:

$$\mathcal{L}_{\text{tie}} = \mathbb{E}_{(i,j) \sim \mathcal{I}} \Big[ \max\{0,\, |u_i - u_j| - \delta\} \Big]. \quad (10)$$

Finally, to control the relative sensitivity of the score to reward versus cost, we regularize the *slope ratio* on a small subsample:

$$g(x) = \frac{\partial f_\omega / \partial r}{-\partial f_\omega / \partial c + \varepsilon},$$
$$\mathcal{L}_{\text{mrs}} = \mathbb{E}_x \Big[ \max\{0,\, \rho_{\min} - g(x)\} \quad (11)$$
$$+ \max\{0,\, g(x) - \rho_{\max}\} \Big],$$
$$0 < \rho_{\min} \leq \rho_{\max}.$$

The final objective blends the three terms:

$$\mathcal{L}_{\text{score}} = w_{\text{dom}} \mathcal{L}_{\text{dom}} + w_{\text{tie}} \mathcal{L}_{\text{tie}} + w_{\text{mrs}} \mathcal{L}_{\text{mrs}}. \quad (12)$$

The margin $m$, tie band $\delta$, and slope-ratio bounds $(\rho_{\min}, \rho_{\max})$ act as weak priors that avoid degenerate solutions (e.g., collapsing scores or extreme reward/cost sensitivity); we use a single default setting across all tasks and report the full hyperparameter setting in the Appendix.

### 3.4. Score Distillation into World Models

To use the learned preference during imagination, a naive approach would query the standalone ScoreNet at every imagined step of every candidate rollout. In model-predictive planning, evaluating $N$ action sequences over horizon $H$ would therefore incur $\mathcal{O}(NH)$ additional forward passes through an auxiliary network (and require feeding it predicted rewards/costs), which unnecessarily increases planning-time compute and coupling between components. We therefore distill ScoreNet into the world model as a native score prediction head that can be evaluated alongside the other world-model heads during rollouts. Crucially, we isolate the dynamics by training the score head on stop-gradient latents, so this preference learning does not backpropagate into the encoder or RSSM.

Let $s_t = (h_t, z_t)$ denote the world-model latent state and $\tilde{s}_t = \text{sg}(s_t)$ its stop-gradient copy. An EMA teacher $\bar{\omega}$

(updated from the ScoreNet parameters $\omega$) produces per-step score targets:

$$y_t^\star = f_{\bar\omega}(r_t, c_t, z_t), \qquad \bar\omega \leftarrow (1-\tau)\bar\omega + \tau\omega, \quad (13)$$

where $(r_t, c_t)$ come from replay. The student score head predicts a heteroscedastic Gaussian distribution:

$$p_\xi(y_t \mid \tilde{s}_t) = \mathcal{N}\Big(\mu_\xi(\tilde{s}_t), \text{diag}(\sigma_\xi^2(\tilde{s}_t))\Big). \quad (14)$$

This setup matches the "missing conditions" nature of distillation: the teacher conditions on $(r_t, c_t, z_t)$, while the student conditions only on $\tilde{s}_t$ and absorbs the uncertainty via $\sigma_\xi$.

We train the score head by minimizing the masked negative log-likelihood:

$$\mathcal{L}_{\text{distill}} = \mathbb{E}_t\left[m_t \cdot \frac{1}{2}\left(\frac{(\mu_{\xi,i}(\tilde{s}_t) - y_{t,i}^\star)^2}{\sigma_{\xi,i}^2(\tilde{s}_t)}\right.\right.$$
$$\left.\left. + \log\sigma_{\xi,i}^2(\tilde{s}_t)\right)\right], \quad (15)$$

where $m_t$ masks valid (non-padding) steps in a sequence batch and ignores terminal/padded steps. For numerical stability, we parameterize $\log\sigma_{\xi,i}^2$ and clamp it into $[\log\sigma_{\min}^2, \log\sigma_{\max}^2]$. Although the teacher ScoreNet is structurally monotone in the explicit reward and cost inputs, the distilled student head is not claimed to be architecturally monotone in the same symbolic sense. This is deliberate: monotonicity is semantically meaningful for the teacher inputs $(r, c, z)$, but the student is queried only from the stop-gradient latent state $\tilde{s}_t$, whose coordinates do not directly correspond to reward or cost. We therefore keep the teacher monotone and train the student to inherit its ordering on the data distribution, while hard safety remains enforced by cost-feasibility gating during planning.

The overall world-model objective augments the standard losses with a weighted distillation term:

$$\mathcal{L}_{\text{WM}} = \mathcal{L}_{\text{recon}} + \mathcal{L}_{\text{reward}} + \mathcal{L}_{\text{cost}} + \mathcal{L}_{\text{cont}} + \lambda_{\text{score}}\mathcal{L}_{\text{distill}}. \quad (16)$$

Here $\mathcal{L}_{\text{recon}}$ denotes the standard world-model reconstruction and dynamics regularization objective (as in SafeDreamer), $\mathcal{L}_{\text{reward}}$ and $\mathcal{L}_{\text{cost}}$ are the reward/cost prediction losses, and $\mathcal{L}_{\text{cont}}$ is the continuation (discount) prediction loss. We linearly warm up $\lambda_{\text{score}}$ from 0 to $\lambda_{\text{score}}^{\max}$ and apply gradient clipping only to the score head for stability.

To reduce the moving-target effect, we update the EMA teacher intermittently: after every $K_{\text{EMA}}$ ScoreNet updates, we refresh $\bar\omega$ using Eq. (13), and then keep $\bar\omega$ fixed for the next $M_{\text{WM}}$ world-model updates so the student does not chase a rapidly drifting teacher. At inference, we default to the mean prediction $\hat{y}_t = \mu_\xi(\tilde{s}_t)$; risk-sensitive variants may use higher quantiles of $p_\xi(y_t \mid \tilde{s}_t)$.

### 3.5. Score-Based Policy Learning and Planning

**ScoreCritic: temporalizing single-step scores.** The distilled score head provides a per-step preference prediction during imagination. To avoid notational clash with the latent state $s_t = (h_t, z_t)$, we denote the predicted single-step score by $\hat{y}_t \triangleq \mu_\xi(\tilde{s}_t)$, where $\tilde{s}_t = \text{sg}(s_t)$. We convert $\{\hat{y}_t\}$ into a long-horizon preference signal using TD($\lambda$), analogous to DreamerV3/SafeDreamer reward and cost critics:

$$G_t^S = \hat{y}_t + \gamma_s\Big((1-\lambda_s)V_{\psi_s}^S(s_{t+1}) + \lambda_s G_{t+1}^S\Big), \quad (17)$$

with standard handling of terminals and truncation masks as in Dreamer/SafeDreamer. We parameterize $V_{\psi_s}^S$ with the same distributional symlog–twohot value head as in DreamerV3/SafeDreamer and train it with the corresponding cross-entropy objective (details in Appendix).

The resulting score advantage is

$$A_t^S = G_t^S - V_{\psi_s}^S(s_t), \quad (18)$$

which is used for actor shaping and for score-based ranking in planning.

**Score-shaped actor update.** USB-RL retains the reward-driven actor objective of DreamerV3/SafeDreamer and uses the learned score only as a lightweight shaping signal. Concretely, we inject a small advantage-based term from the ScoreCritic into the standard imagination-based actor update:

$$L_{\text{actor}}(\theta) = -\sum_{t=0}^{T-1}\text{sg}\big(R_t^\lambda + \alpha_{\text{score}}A_t^S\big) + \eta\sum_{t=0}^{T-1}\mathcal{H}\big(\pi_\theta(\cdot|s_t)\big),$$

where $R_t^\lambda$ is the usual reward TD($\lambda$) target in DreamerV3/SafeDreamer and $\alpha_{\text{score}}$ is fixed across tasks. Using the score *advantage* (rather than raw score returns) makes the shaping baseline-invariant and stabilizes learning. Importantly, this term does not replace the reward objective and does not relax safety constraints: hard cost-feasibility is enforced by the planner, and the shaping term merely biases the actor toward behaviors preferred by the learned score within the feasible set.

**Safe score-ranked planning (OSRP).** To handle CMDP constraints, we *retain* OSRP's strict safety gating and use the learned score *only* to rank within cost-feasible candidates. At each decision step, the planner samples $N$ candidate action sequences and rolls them out for horizon $H$ in the world model, yielding imagined trajectories $\{\tau_i\}_{i=1}^N$. For each trajectory $\tau_i$, we compute the predicted cost return (as in SafeDreamer) $\hat{J}_\phi^C(\tau_i) = \sum_{t=0}^{H-1}\gamma^t\hat{c}_\phi(z_t, a_t)$ and the score return $\hat{J}^S(\tau_i) = G_0^S(\tau_i)$ using Eq. (17). We keep only cost-feasible candidates:

$$\mathcal{T}_{\text{safe}} = \{\tau_i \mid \hat{J}_\phi^C(\tau_i) \leq B\}, \quad (19)$$

where $B$ is a task-level cost budget (typically with a small slack in model-based planning). If $|\mathcal{T}_{\text{safe}}| < N_s$ (safety mode), we pick the candidate with the smallest predicted cost; otherwise (performance mode), we pick the score-best candidate within $\mathcal{T}_{\text{safe}}$:

$$\tau^\star = \begin{cases} \arg\min_{\tau_i} \hat{J}^C_\phi(\tau_i), & |\mathcal{T}_{\text{safe}}| < N_s, \\ \arg\max_{\tau \in \mathcal{T}_{\text{safe}}} \hat{J}^S(\tau), & \text{otherwise.} \end{cases} \quad (20)$$

The executed action is the first action of $\tau^\star$. The flexibility to switch between safety and performance modes based on available candidates is what enables our approach to effectively balance risk and reward. We follow OSRP's adaptive planning strategy from SafeDreamer and use the learned score to rank cost-feasible candidates. These parameters are adjusted dynamically, with shorter horizons and increased safety margins used when uncertainty is high, and longer horizons with more extensive exploration allowed when the system is more predictable.

## 4. Experiments

### 4.1. Setup

**Benchmarks.** We validate the effectiveness of USB-RL in handling reward–cost conflicts on two classical safety-driven visual RL benchmarks: **Safety-Gymnasium** and **MetaDrive**. We adopt the same task families and evaluation protocol as SafeDreamer, and use each environment's default specifications for cost signals and budgets. Additional details are provided in the supplementary materials.

**Compared methods.** We compare against: (1) **Safe-Dreamer** (Huang et al., 2024), OSRP world-model planning with cost critics and Lagrangian updates; (2) **PPO-Lag** (Ray et al., 2019), a strong model-free CMDP baseline; (3) **LAMBDA** (As et al., 2022), implemented in Dreamer-V1, combines Bayesian and Lagrangian methods. On MetaDrive we additionally report **PPO-RS**, a PPO baseline with reward shaping that incorporates the safety cost as a shaping term rather than as an explicit constraint. For fairness, dynamics, encoders, actor/value heads, and training infrastructure are kept identical across Dreamer-style methods; only the objective (penalty vs. USB-RL score) and the extra score heads differ.

### 4.2. Results on Safety-Gymnasium

Fig. 3 summarizes the learning curves on four Safety-Gymnasium tasks (PointGoal2, PointButton1, PointPush1, CarGoal1). For each method, we plot the average episode reward (top) and average episode cost (bottom) over training steps, averaged over 5 seeds with shaded 95% confidence intervals. Across all tasks, USB-RL achieves a consistently better safety–performance trade-off than existing safe RL baselines. On **PointGoal2** and **PointButton1**, USB-RL attains higher asymptotic reward than both LAMBDA and

SafeDreamer, while driving the cost substantially below the baselines, often approaching near-zero violations. On the more challenging **PointPush1**, LAMBDA can reach relatively high reward but does so at noticeably higher cost, whereas USB-RL matches or exceeds the reward of Safe-Dreamer while achieving the lowest cost among safe methods. This indicates that learning a unified, structured score is particularly beneficial when reward and cost are strongly coupled: it provides a stable ranking signal for nuanced trade-offs that are difficult to capture with a single linear penalty. On **CarGoal1**, which combines long-horizon planning with tight safety constraints, USB-RL again provides the best trade-off: it achieves higher reward than LAMBDA and SafeDreamer while substantially reducing cost, approaching the lowest cost across methods. Overall, these results indicate that USB-RL can leverage world models to recover strong returns while substantially reducing safety violations, consistently dominating or matching prior safe RL baselines in both reward and cost across all tested tasks.

### 4.3. Score Manifold Analysis

To examine whether the learned score induces a meaningful ordering beyond linear cost penalties, we visualize the joint relation between score, reward, and cost. At each evaluation checkpoint during training (the same data used for the learning curves), we record the average episode reward and average episode cost, and pair them with the corresponding score statistic (Fig. 6). The resulting point cloud forms a curved, non-planar manifold: high scores concentrate in the low-cost region while still spanning a range of rewards, whereas low-score points exhibit substantially higher costs and lower rewards. The thickness of the manifold suggests context-dependent trade-offs, consistent with the score depending on the latent context $z$, and indicates that the learned score is not a mere linear scalarization of reward and cost. Overall, this analysis supports that USB-RL learns a globally consistent preference ordering while accommodating non-linear, state-dependent safety–performance trade-offs.

### 4.4. Results on MetaDrive

We further evaluate USB-RL on the MetaDrive driving benchmark. Following the evaluation protocol of Safe-Dreamer, we report (i) *arrival rate* (higher is better) and (ii) *episode cost return* (lower is better) after $4 \times 10^6$ environment steps. In MetaDrive, the per-step safety cost is a sparse binary signal that fires upon unsafe events (e.g., collision or leaving the road), and the episode cost return aggregates these events over an episode. As shown in Fig. 4, USB-RL attains the highest arrival rate among all methods, clearly outperforming SafeDreamer and PPO-based baselines. At the same time, USB-RL achieves the lowest episode cost return, indicating substantially fewer safety violations at a comparable (or better) level of task completion. These re-

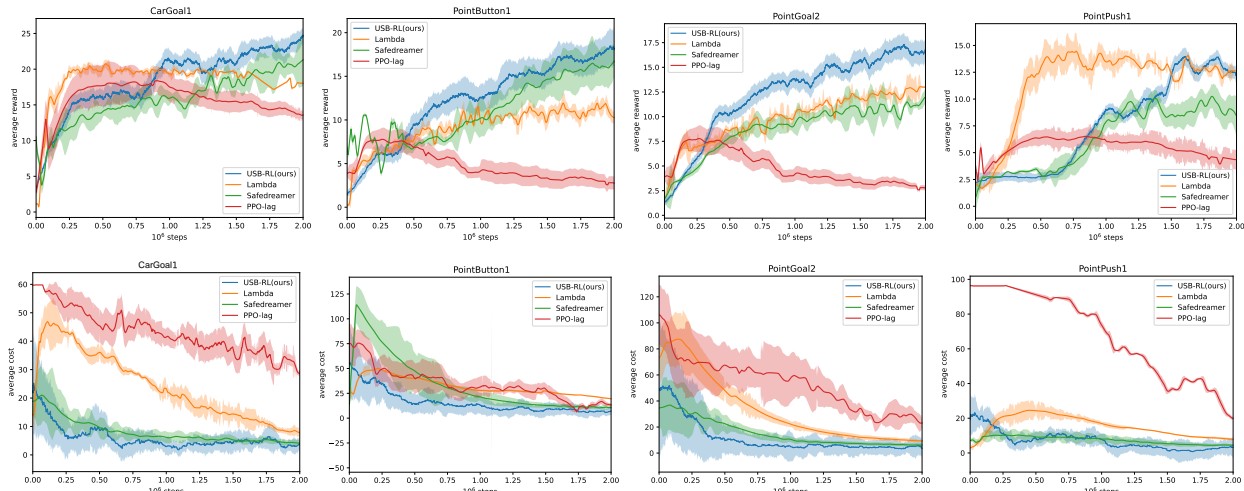

*Figure 3. Results on Safety-Gymnasium.* USB-RL attains higher returns (top) with fewer violations (bottom). Shaded areas indicate the 95% confidence interval over 5 seeds.

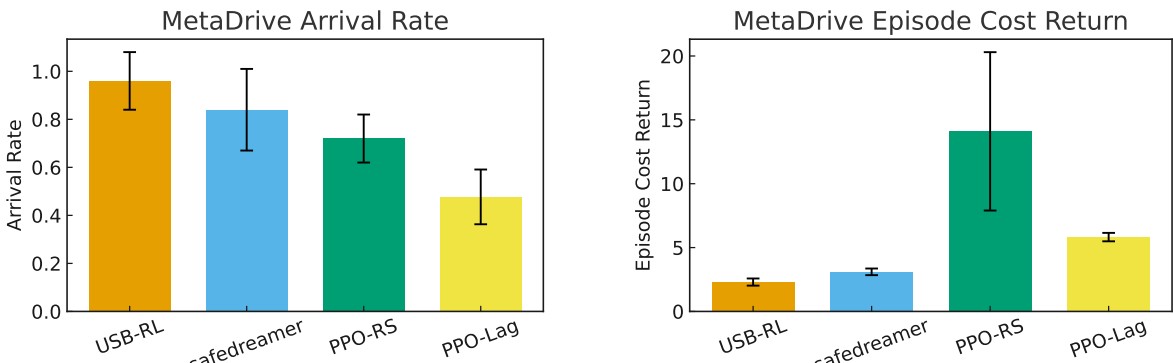

*Figure 4. Results on MetaDrive.* Final arrival rate (left, higher is better) and episode cost return (right, lower is better) on the MetaDrive benchmark. Error bars denote the standard deviation across random seeds.

sults suggest that the learned unified score transfers beyond Safety-Gymnasium and scales to realistic driving scenes, where reward-cost trade-offs can be highly non-linear and state-dependent.

### 4.5. Ablation Studies

To understand the contribution of each component of USB-RL, we conduct targeted ablations on the PointButton1 task. We compare four variants: (**Full**) the complete USB-RL algorithm; (**No-Distill**) removing score distillation into the world model; (**No-Pair**) replacing the within-root dominance/tie pair construction for ScoreNet with uniformly random pairwise comparisons sampled from a mixed pool of replay and imagined summaries; and (**No-ScoreCritic**) removing temporal score propagation during imagination. As shown in Fig. 5, each component contributes meaningfully.

**Effect of distillation.** Removing distillation retains the learned teacher ScoreNet but prevents its preference from being embedded as a native world-model head. The planner must instead query the external ScoreNet when evaluating imagined rollouts, which increases coupling between com-

ponents and makes the score signal less tightly aligned with the latent dynamics used by imagination. As a result, No-Distill can remain competitive early in training but lags behind the full method in the long run, particularly in reward. This gap does not imply that the distilled student is a better score model than the teacher in isolation; rather, it shows the benefit of integrating the score into imagination-time control. The stop-gradient distilled head provides a native score signal that is available alongside the reward and cost heads during rollouts and is consistently used by both planning and the ScoreCritic.

**Effect of structured pair construction.** The No-Pair variant removes the within-root pair construction (Sec. 3.3) used to define informative dominance/tie relations, and instead relies on uniformly random pairs drawn from a joint pool of replay and imagination summaries. This substantially reduces the frequency of informative comparisons near the reward-cost trade-off frontier, where ordering signals are most valuable but also most ambiguous. Consequently, its reward curve grows more slowly and saturates below the full method, while its cost curve shows higher variance and slower convergence. This aligns with the intuition that, with-

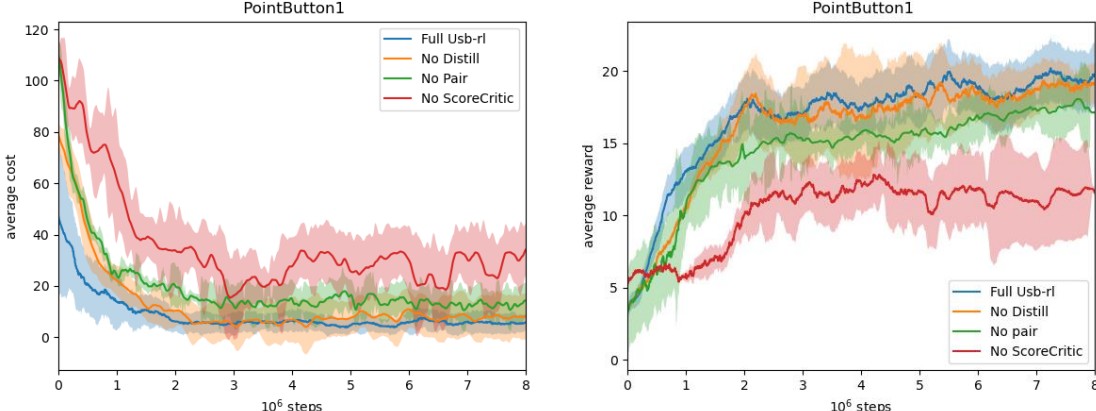

Figure 5. *Ablation studies.* All three components of our method contribute effectively to performance.

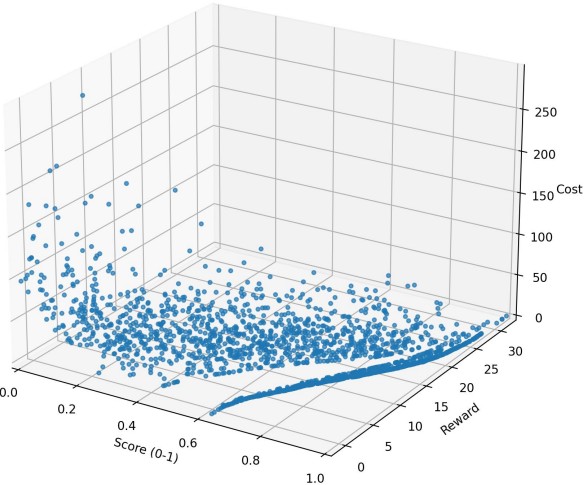

Figure 6. *Reward–cost–score manifold.* 3D scatter over evaluation checkpoints during training, plotting (score, average episode reward, average episode cost). The curved, non-planar structure suggests non-linear safety–performance trade-offs beyond linear penalties.

out structured comparisons, ScoreNet learns a noisier and less consistent preference signal, which is less effective for guiding score-ranked planning.

**Effect of the ScoreCritic.** The No-ScoreCritic variant (removing temporal score propagation) produces the most pronounced degradation. Without a value function over scores, the planner only receives one-step preferences and cannot reason about long-horizon trade-offs, leading to lower reward and higher cost throughout training. This highlights that temporalizing the score signal is a key mechanism through which USB-RL achieves reliable control.

## 5. Related work

**Safe RL and CMDP.** Safe reinforcement learning (Safe RL) is typically formalized as a constrained Markov decision process (CMDP) (Altman, 1999). Classical approaches include primal–dual updates that treat the Lagrange multiplier as a learnable parameter (Chow et al., 2018; Tessler et al., 2019; Stooke et al., 2020), trust-region methods with explicit constraints such as CPO/PCPO (Achiam et al., 2017; Yang et al., 2020), and control-theoretic ideas like Lyapunov critics, barrier functions, or shielding (Chow et al., 2018; Dalal et al., 2018; Könighofer et al., 2022). Recent works (2022–2025) refine these pathways with adaptive multipliers (Ray et al., 2019), budgeted/value-decomposition perspectives (Efroni et al., 2021), and updated surveys emphasizing evaluation protocols and multi-constraint settings (Thomas et al., 2021). A recurrent challenge across these lines is scalarization and ambiguous rankings when trajectories trade off safety and performance near budgets. This motivates representations that preserve dominance and recognize incomparability rather than flattening them into a single hand-tuned scalar.

**Model-based visual RL.** World models learn compact latent dynamics and optimize policies using imagined rollouts (Ha & Schmidhuber, 2018; Hafner et al., 2020b;a). Dreamer-style RSSMs pair latent imagination with distributional value heads and actor updates, yielding strong sample efficiency and broad applicability. DreamerV3 further standardized robust training across 150+ tasks with balanced losses, symlog/two-hot parameterizations, adaptive normalization, and regularization that stabilizes long-horizon imagination (Hafner et al., 2023). Subsequent variants extend planning horizons, uncertainty handling, or decision-time guidance while keeping the core idea: optimize in imagination with a model the agent trains itself (Janner et al., 2019; Schrittwieser et al., 2020; Hansen et al., 2022).

**Model-based safe RL.** Bringing safety into model-based RL raises two coupled issues: (i) how to propagate and prioritize safety signals during imagination, and (ii) how to avoid runtime overhead from external safety modules. Early model-based safe RL combined learned dynamics with constrained policy optimization or Lagrangian penalties (Jayant & Bhatnagar, 2022; Könighofer et al., 2022); others incorporated conservative or risk-sensitive planning objectives (Zhang et al., 2024). SafeDreamer integrates safety directly into Dreamer-style control by performing online safe planning within the world model (Huang et al., 2024). Concretely, OSRP/OSRP-Lag apply constrained cross-entropy search over imagined rollouts with cost critics for gating, while BSRP-Lag amortizes planning with background policy updates. Empirically, SafeDreamer shows that vision-based agents can satisfy constraints with manageable compute by letting the world model propose and filter candidates. Concurrently, several works explore deeper integration of safety with learned models: safety-aware trajectory optimization that adapts feasible sets during imagination (Yang et al., 2023), budgeted MDPs that condition policies on adjustable cost budgets (Carrara et al., 2019), and methods that exploit privileged cost information during training to improve safe representations (Huang et al., 2025). These directions share a theme with our work, but largely continue to rely on penalty schedules or supervised preferences.

USB-RL sits at the intersection of safe RL, world models, and preference learning. Our approach is complementary to SafeDreamer-style planners and can be viewed as a drop-in augmentation that equips imagination with a coherent, safety–performance ordering.

## 6. Conclusions and Limitations

In this work, we proposed USB-RL, a model-based safe reinforcement learning framework that integrates a learned, structured safety–performance score directly into imagination-based planning. USB-RL differs from penalty-based methods by avoiding hand-tuned reward–cost penalty scalarization, from supervised preference learning by constructing self-supervised preference signals from reward–cost outcomes without human preference labels, and from two-stage pipelines by tightly combining score learning with imagination through distillation and a ScoreCritic. The learned score is first represented by a structurally monotone teacher ScoreNet, then distilled into the world model as a native score head, and finally used for score-ranked planning under cost-feasibility gating. Experiments on standard visual safe-RL benchmarks show that USB-RL improves returns while reducing safety violations, yielding stable safety–performance trade-offs.

USB-RL also has several limitations. It does not provide formal hard-safety guarantees; its safety behavior depends on the accuracy of the learned world model, reward and cost predictors, and finite-horizon planner. Model error, inaccurate cost prediction, or distribution shift can still lead to unsafe actions. In addition, although the teacher ScoreNet is structurally monotone in the explicit reward and cost inputs, the distilled student head is only an empirical surrogate of the teacher-induced ordering over latent states, not a formally monotone function over latent coordinates. The learned ordering also depends on the dominance/tie thresholds and on the informativeness of the sampled within-root comparisons. In safety-critical applications, USB-RL should therefore be combined with verification, robust planning, or additional constraint-enforcement mechanisms.

Finally, USB-RL is not inherently tied to visual observations. The core components—score learning, score distillation, the ScoreCritic, and safe score-ranked planning—operate on world-model latents rather than raw pixels, so the main modality-specific component is the observation encoder. In principle, the framework can be applied to other observation modalities, including low-dimensional state inputs, as long as the world model provides latent representations suitable for imagination and reward/cost prediction. Our empirical claims are nonetheless restricted to the evaluated visual world-model setting. Extending USB-RL to broader control settings and more robust planning under model uncertainty remains important future work.

## Acknowledgements

This work was supported by the National Natural Science Foundation of China (No. 42201513), the China Post-doctoral Science Foundation (Nos. 2022M723902 and 2023T160789) and the Smart Grid National Science and Technology Major Project (No. 2024ZD0801200).

## Impact Statement

This paper presents work whose goal is to advance the field of Machine Learning. There are many potential societal consequences of our work, none which we feel must be specifically highlighted here.

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

---

**Algorithm 1** USB-RL: Model-based Safe RL with Unsupervised Score Learning

---

**Require:** Environment $\mathcal{E}$, cost budget $B$, configuration cfg
1: Initialize world model $\theta$, reward/cost critics $\phi^R, \phi^C$, actor $\theta_\pi$, score head $\eta^S$ and ScoreCritic head $\eta^V$
2: Initialize ScoreNet parameters $\omega$ and EMA teacher $\bar{\omega} \leftarrow \omega$
3: Initialize replay buffer $\mathcal{D} \leftarrow \emptyset$
4: **for** training step $k = 1, \ldots, K$ **do**
5:     **for** $t = 1, \ldots, H_{\text{env}}$ **do**
6:         Encode $s_t = (h_t, z_t) \leftarrow \text{Encode}_\theta(o_t)$
7:         **if** warmup **then**
8:             Sample $a_t \sim \text{Unif}(\mathcal{A})$
9:         **else**
10:             $a_t \leftarrow \text{SafePlanner}(s_t, \theta, \phi^R, \phi^C, \eta^S, \eta^V, B)$
11:         **end if**
12:         Step env: $(o_{t+1}, r_t, c_t, d_t) \sim \mathcal{E}(a_t)$ , store in $\mathcal{D}$
13:         **if** $d_t$ **then**
14:             Reset environment and **break**
15:         **end if**
16:     **end for**
17:     **for** $u = 1, \ldots, U_{\text{wm}}$ **do**
18:         Sample real batch $B_{\text{real}}$ from $\mathcal{D}$
19:         Update world model $\theta$ on reconstruction and KL losses (as in SafeDreamer)
20:         Imagine rollouts from $q_\theta(s_0 \mid B_{\text{real}})$ under $p_\theta$
21:         Update reward/cost critics $\phi^R, \phi^C$ using TD($\lambda$) returns
22:         Compute reward advantages $A_t^R$ from $R_t^\lambda$ and $V_{\phi^R}^R$ (Sec. 3.5)
23:         Update ScoreCritic $\eta^V$ and obtain score advantages $A_t^S$ (Alg. 3)
24:         Update actor $\theta_\pi$ from imagined rollouts using the reward-driven objective with potential-based score shaping
25:     **end for**
26:     **if** $k \bmod K_{\text{score}} = 0$ **then**
27:         Sample real batch $B_{\text{real}}$ and imagined batch $B_{\text{imag}}$
28:         Compute long-horizon TD($\lambda$) returns $G^R, G^C$ with EMA critics (Sec. 3.2)
29:         $\text{UpdateScoreNet}(\omega, \bar{\omega}, B_{\text{real}}, B_{\text{imag}}, G^R, G^C)$ (Alg. 2)
30:         Distill $f_{\bar{\omega}}(r_t, c_t, z_t)$ into the world-model score head $\eta^S$ using the distributional regression loss in Sec. 3.4, with stop-gradient latents
31:     **end if**
32: **end for**

---

## A. Details of USB-RL

**Cost Threshold Settings.** Following SafeDreamer (Huang et al., 2024), we use a fixed cost budget $B$ to gate candidate trajectories during world model planning. Concretely, for each imagined rollout $\tau = (z_0, a_0, \ldots, z_H)$ we compute the $H$-step discounted cost return under the learned cost head,

$$J_\phi^C(\tau) = \sum_{t=0}^{H-1} \gamma^t \hat{c}_\phi(z_t, a_t), \tag{1}$$

and retain only those trajectories whose predicted cost stays below a task-level budget, $J_\phi^C(\tau) \leq B$. In all Safety-Gymnasium visual control tasks, we set a common budget $B = 2.0$.

This choice follows the empirical observation in SafeDreamer that, due to approximation errors in the learned cost and cost-value models, the predicted cost return can remain slightly positive even for essentially safe trajectories. Using a strictly zero cost threshold would therefore cause the planner to reject almost all candidates and become overly conservative, which prevents convergence to high-return policies. A small positive slack $B = 2.0$ instead yields a robust trade-off: it keeps the planner focused on low-cost trajectories while still providing enough feasible candidates for meaningful score-based ranking. We adopt the same convention throughout our Safety-Gymnasium experiments and do not tune $B$ per environment.

---

**Algorithm 2** UpdateScoreNet: Structured Monotone Score Learning

---

**Require:** Student parameters $\omega$, EMA teacher $\bar{\omega}$
**Require:** Real batch $B_{\text{real}}$, imagined batch $B_{\text{imag}}$
1: Build a mixed set of trajectory summaries $\{(x_i, G_i^r, G_i^c)\}_{i=1}^N$ from $B_{\text{real}}$ and $B_{\text{imag}}$, where $x_i = (r_i, c_i, z_i)$ and $(G_i^r, G_i^c)$ come from TD($\lambda$) with EMA reward/cost critics
2: Sample $K$ index pairs $(i, j)$ per summary and compute $\Delta G_{ij}^r, \Delta G_{ij}^c$
3: Construct dominance set $\mathcal{D}$ and incomparable set $\mathcal{I}$ using thresholds $\varepsilon_r, \varepsilon_c$ and hard mining
4: Compute scores $s_i = f_\omega(x_i)$ with the monotone ScoreNet
5: Dominance loss: $\mathcal{L}_{\text{dom}} = \mathbb{E}_{(i,j)\in\mathcal{D}}[\max\{0,\ m - (s_i - s_j)\}]$
6: Tie loss: $\mathcal{L}_{\text{tie}} = \mathbb{E}_{(i,j)\in\mathcal{I}}[\max\{0,\ |s_i - s_j| - \delta\}]$
7: On a subsample $\mathcal{S}$, compute slope ratios $g(x) = (\partial f_\omega/\partial r)/(-\partial f_\omega/\partial c)$ and the monotone-ratio loss $\mathcal{L}_{\text{mrs}}$
8: Total loss: $\mathcal{L}_{\text{score}} = w_{\text{dom}}\mathcal{L}_{\text{dom}} + w_{\text{tie}}\mathcal{L}_{\text{tie}} + w_{\text{mrs}}\mathcal{L}_{\text{mrs}}$
9: Take a gradient step on $\omega$ to minimize $\mathcal{L}_{\text{score}}$
10: Apply row-wise Lipschitz projection to positive weights in ScoreNet
11: Update EMA teacher: $\bar{\omega} \leftarrow (1 - \tau)\bar{\omega} + \tau\omega$
12: **return** Updated $(\omega, \bar{\omega})$

---

**Algorithm 3** ScoreCritic Update inside Imagination

---

**Require:** Latent rollouts $\{z_t\}_{t=0}^{T-1}$, score head $\eta^S$, ScoreCritic head $\eta^V$
1: Compute single-step scores $\hat{s}_t = S_{\eta^S}(z_t)$
2: Compute TD($\lambda$) targets $G_t^S$ from $\{\hat{s}_t\}$ using discount $\gamma$ and trace $\lambda$ (shared with reward/cost critics)
3: Update distributional value head $V_{\eta^V}^S$ with the same symlog two-hot loss as SafeDreamer, minimizing the discrepancy between $V^S(z_t)$ and $G_t^S$
4: **return** Updated $\eta^V$

---

**Actor Training.** The actor in USB-RL follows the reward-driven design of Dreamer and SafeDreamer and is trained purely from imagined trajectories under the learned world model. Let $s_t = (h_t, z_t)$ denote the latent state, $a_t \sim \pi_\theta(\cdot \mid s_t)$ the action sampled from the stochastic policy, and $R^\lambda(s_t)$ the TD($\lambda$) reward return computed with the same $(\gamma, \lambda)$ as in SafeDreamer. Following SafeDreamer, the base actor objective balances the maximization of expected reward and entropy:

$$\mathcal{L}_{\text{actor}}(\theta) = -\sum_{t=0}^{T-1} \text{sg}\left(R^\lambda(s_t)\right) + \eta \sum_{t=0}^{T-1} \text{H}\left[\pi_\theta(a_t \mid s_t)\right] \tag{2}$$

where $\eta$ is an entropy coefficient and $\text{sg}(\cdot)$ denotes stop-gradient with respect to the value networks. As in Dreamer and SafeDreamer, gradients of the reward term are estimated by stochastic backpropagation through the reparameterized policy and world model, while the entropy term admits an analytic gradient; $\text{sg}(\cdot)$ only indicates that critic parameters are not updated by this loss.

USB-RL augments this reward-driven actor with a shaping signal derived from the ScoreCritic. Given the temporal score signal $G_t^S$ and its value estimate $V^S(z_t)$ (implemented as a symlog two-hot head as in SafeDreamer), we define the score advantage

$$A_t^S = G_t^S - V^S(z_t). \tag{3}$$

We then modify the reward term by injecting $A_t^S$ with a small coefficient $\alpha_{\text{score}}$ that shares the same gradient path as the reward return:

$$\mathcal{L}_{\text{actor}}^{\text{USB-RL}}(\theta) = -\sum_{t=0}^{T-1} \text{sg}\left(R^\lambda(s_t) + \alpha_{\text{score}} A_t^S\right)$$
$$+ \eta \sum_{t=0}^{T-1} \text{H}\left[\pi_\theta(a_t \mid s_t)\right]. \tag{4}$$

In all experiments, $\alpha_{\text{score}}$ is set to a small constant and is kept fixed across tasks. Because $A_t^S$ is derived from a value function $V^S$ and enters only through the same reparameterization path as the reward return, this score-based shaping acts as

a potential-like adjustment to the reward signal, providing a more informative unified safety–performance learning signal. Empirically we observe that enabling this shaping term ($\alpha_{\text{score}} > 0$) accelerates reward convergence and helps maintain low violation rates, as shown in the learning curves in the main paper.

**Reward, cost, and score critics.** USB-RL follows SafeDreamer in how it parameterizes reward and cost value functions, and extends the same design to the ScoreCritic. Let $R_t^\lambda$ and $C_t^\lambda$ denote the TD($\lambda$) returns for reward and cost, computed with the same $(\gamma, \lambda)$ as in the main agents. Each critic head outputs a categorical distribution over a fixed bin set $B = [b_{\min}, \ldots, b_{\max}]$. Targets are obtained by applying the bi-symmetric log transform $\text{symlog}(x)$ and encoding the result as a twohot vector over $B$, as in DreamerV3. The reward critic with parameters $\psi_r$ is trained by minimizing the cross-entropy between the predicted logits and the two-hot-encoded symlog target, *i.e.*, the loss $\mathcal{L}_{\text{reward-critic}}(\psi_r)$ from SafeDreamer, and the cost critic loss $\mathcal{L}_{\text{cost-critic}}(\psi_c)$ is defined analogously using $C_t^\lambda$. Scalar values are recovered via the inverse transform, $v_\psi(s_t) = \text{symexp}\big(p_\psi(\cdot \mid s_t)^\top B\big)$.

The ScoreCritic reuses the same symlog–twohot parameterization. Given single-step scores $s_t$ from the world model score head and the TD($\lambda$) score returns $G_t^S$ defined, we train a score value head $V^S(z_t)$ with the MSE objective, $\mathcal{L}_{\text{score-critic}} = \mathbb{E}_t[(V^S(z_t) - G_t^S)^2]$. Here we use the same discount and trace parameters $(\gamma_s, \lambda_s)$ as for the reward and cost critics, and obtain $V^S(z_t)$ by decoding the symlog twohot distribution in exactly the same way as for reward and cost. The resulting score advantage $A_t^S = G_t^S - V^S(z_t)$ is then shared with the actor and the safe planner to provide a unified temporal safety–performance signal.

# B. Hyperparameters

The experiments were executed utilizing Python3 and Jax 0.6.2, facilitated by CUDA 12.2, on an Ubuntu 20.04.5 LTS system equipped with Intel(R) Xeon(R) Gold 6330 CPU @ 2.00GHz, and an array of 4 GeForce RTX 4090Ti GPUs.

# C. Experiments

## C.1. Experiments on Safety-Gymnasium

### C.1.1. WALL-CLOCK OVERHEAD OF SCORE DISTILLATION.

Table 5 reports the wall-clock training time of USB-RL with and without the score distillation module. The variant without distillation removes the world-model score head and trains only the external ScoreNet, while the full USB-RL additionally distills the learned scores into the world model and trains the ScoreCritic. We observe that the full method is faster than the no-distill variant because the distilled score head avoids repeatedly querying the external ScoreNet during planning. This is consistent with our design: ScoreNet is updated at a low frequency (every $K_{\text{score}}$ environment steps) and distillation only affects a small additional head on top of the world model, so the computational overhead is limited, whereas the benefits in stability and safety–performance trade-offs are substantial (cf. the main learning curves and violation rates).

Safety-Gymnasium is an environment library specifically designed for SafeRL. This library builds on the foundational `Gymnasium` API, utilizing the high-performance MuJoCo engine. We conducted experiments in four different environments, namely, `SafetyPointGoal2`, `SafetyPointPush1`, `SafetyPointButton1`, and `SafetyCarGoal1`, as illustrated in Figure 1. Following safedreamer (Huang et al., 2024), we adjusted the arrangement of the cameras and used both the forward and rear views to provide more information to planning-based algorithms for processing visual inputs compared to Ray et al. (2019). We use the same environments as SafeDreamer and apply identical modifications to the state representation to better facilitate model learning. All experiments are conducted under identical configuration settings.

### C.1.2. AGENT

We consider three robots: Point, Car and Racecar (as shown in Figure. 2). To potentially enhance learning with neural networks, we maintain all actions as continuous and linearly scaled to the range of [-1, +1]. Detailed descriptions of the robots are provided as follows:

**Point**. The Point robot, functioning in a 2D plane, is controlled by two distinct actuators: one governing rotation and another controlling linear movement. This separation of controls significantly eases its navigation. A small square, positioned in the robot's front, assists in visually determining its orientation and crucially supports Point in effectively manipulating boxes encountered during tasks.

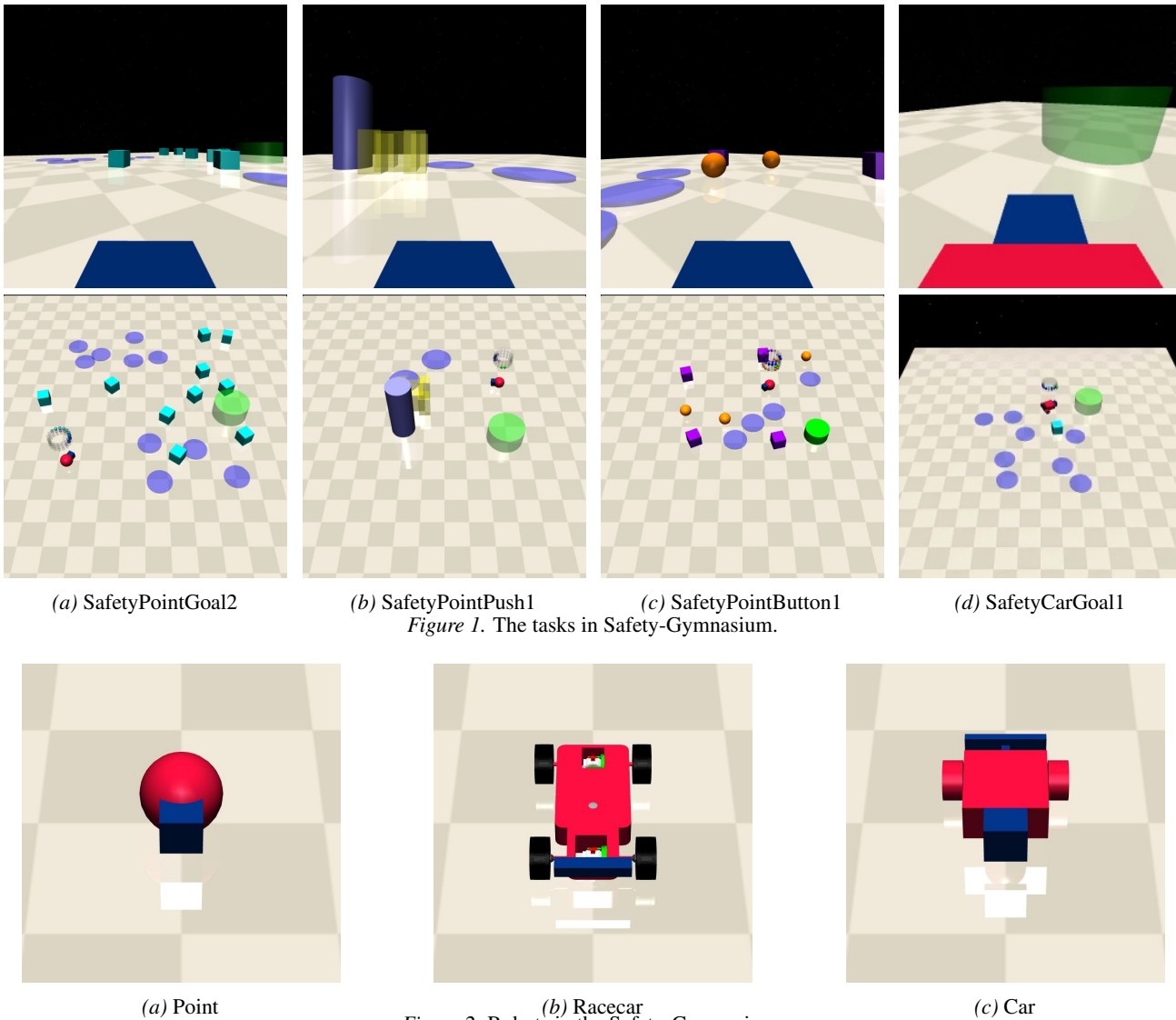

*(a)* SafetyPointGoal2          *(b)* SafetyPointPush1          *(c)* SafetyPointButton1          *(d)* SafetyCarGoal1

*Figure 1.* The tasks in Safety-Gymnasium.

*(a)* Point          *(b)* Racecar          *(c)* Car

*Figure 2.* Robots in the Safety-Gymnasium.

**Car**. The Car robot apparatus operating within a three-dimensional space. It is equipped with two independently powered wheels positioned in parallel, complemented by a rear wheel that rolls freely. This configuration necessitates a coordinated manipulation of the dual propulsion systems to achieve both navigational steering and linear movement in the forward and reverse directions. Although this robot exhibits characteristics akin to those of a rudimentary Point robot, it introduces additional complexities due to its design.

**Racecar**. The Racecar robot exhibits realistic car dynamics, operating in three dimensions, and controlled by a velocity and a position servo. The former adjusts the rear wheel speed to the target, and the latter fine-tunes the front wheel steering angle. The dynamics model is informed by the widely recognized MIT Racecar project. To achieve the designated goal, it must appropriately coordinate the steering angle and speed, mirroring human car operation.

### C.1.3. TASK

Tasks within Safety-Gymnasium are distinct and are confined to a single environment each, as shown in Figure 3.

**Goal**: The task requires a robot to navigate towards multiple target positions. Upon each successful arrival, the robot's goal position is randomly reset, retaining the global configuration. Attaining a target location, signified by entering the goal circle, provides a sparse reward. Additionally, a dense reward encourages the robot's progression through proximity to the target.

**Push**: The task requires a robot to manipulate a box towards several target locations. Like the goal task, a new random target

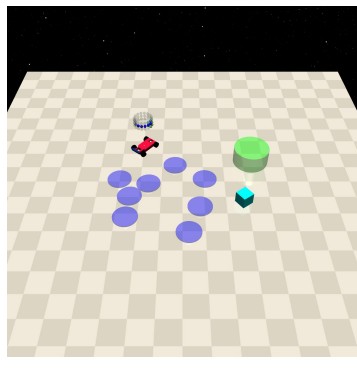

*(a)* Goal

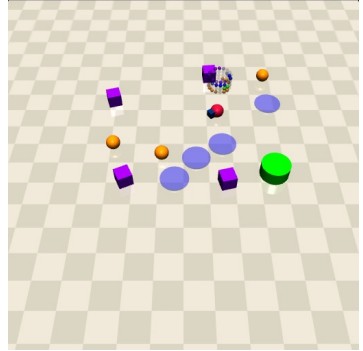

*(b)* Button

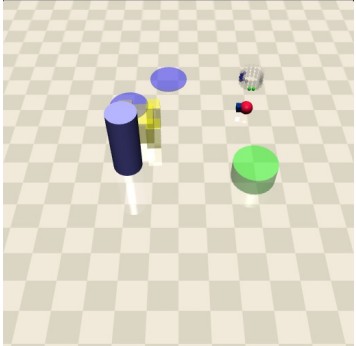

*(c)* Push

*Figure 3.* Task types in the Safety-Gymnasium.

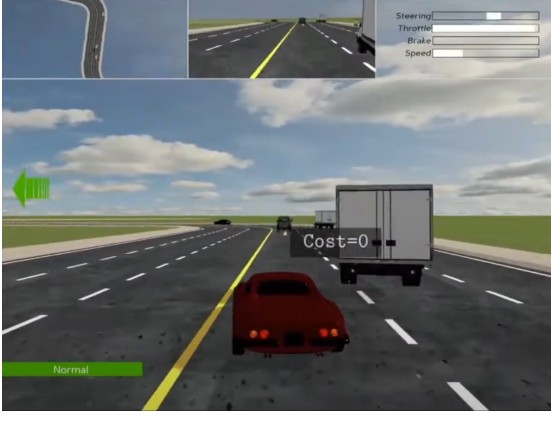

*Figure 4.* Safe

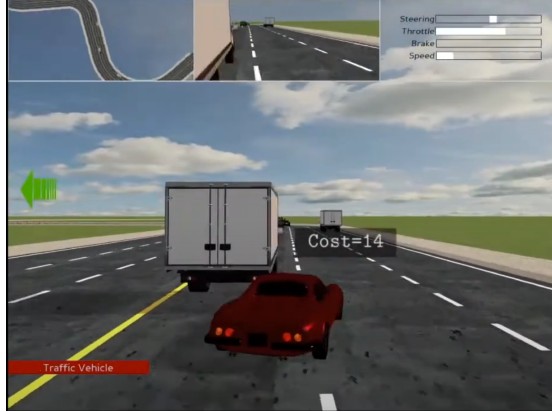

*Figure 5.* Unsafe

*Figure 6.* The MetaDrive benchmark. The objective for the car is to navigate successfully to a predetermined destination. During this process, the vehicle incurs a cost penalty in instances of collision with obstacles or other vehicles, as well as when deviating from the designated roadway.

location is generated after each successful completion. The sparse reward is granted when the box enters the designated goal circle. The dense reward comprises two parts: one for narrowing the agent-box distance, and another for advancing the box towards the final target.

**Button**: The task requires the activation of numerous target buttons distributed across the environment. The agent navigates and interacts with the currently highlighted button, the goal button. Upon pressing the correct button, a new goal button is highlighted, while maintaining the overall environment. The sparse reward is issued upon successfully activating the current goal button, with the dense reward component encouraging progression toward the highlighted target button.

### C.2. Metadrive

MetaDrive stands as a comprehensive, effective simulation environment for autonomous vehicle research. It features designed environments optimized for developing safe policies. The observation in MetaDrive combines vector input and first-person view input. The cost function in this setting is defined as follows:

$$C(s, a) = \begin{cases} 1, & \text{if collides or out of the road} \\ 0, & \text{otherwise} \end{cases} \tag{5}$$

We maintained a consistent environment by conducting both training and testing on the identical roadway. To augment the complexity, vehicle positions were randomized at the beginning of each episode reset.

*Table 1.* Hyperparameters for USB-RL. When addressing other safety tasks, we suggest tuning the initial Lagrangian multiplier, proportional coefficient, integral coefficient, and difference coefficient at various scales.

| Name | Symbol | Value | Description |
|---|---|---|---|
| **ScoreNet** | | | |
| ScoreNet hidden units | – | 64 | – |
| ScoreNet depth | – | 2 | – |
| Row-sum Lipschitz bound | $L_{\max}$ | 3.0 | – |
| Update interval | $K_{\mathrm{score}}$ | 10 000 | env steps |
| ScoreNet learning rate | – | $10^{-4}$ | – |
| EMA coefficient | $\tau$ | 0.005 | – |
| Pairs per sample | $K$ | 4 | – |
| Dominance thresholds | $\varepsilon_r, \varepsilon_c$ | $0.2 \to 0.05$ | – |
| Hinge margin | $m$ | 0.1 | – |
| Tie band | $\delta$ | 0.05 | – |
| Slope ratio bounds | $\rho_{\min}, \rho_{\max}$ | 0.5, 2.0 | – |
| MRS subsample rate | – | 0.05 | – |
| Loss weights | $w_{\mathrm{dom}}, w_{\mathrm{tie}}, w_{\mathrm{mrs}}$ | 1.0, 0.5, 0.1 | – |
| **Score distillation** | | | |
| Max distill weight | $\lambda_{\mathrm{score}}^{\max}$ | 0.2 | – |
| Warmup steps | – | $5 \cdot 10^4$ | env steps |
| **ScoreCritic** | | | |
| Score discount | $\gamma_s$ | 0.997 | – |
| Score trace | $\lambda_s$ | 0.95 | – |
| **World Model** | | | |
| Number of latent units | – | 48 | – |
| Classes per latent | – | 48 | – |
| Batch size | $B$ | 64 | – |
| Batch length | $T$ | 16 | – |
| Learning rate | – | $10^{-4}$ | – |
| KL coeff. | $\alpha_q, \alpha_p$ | 0.1, 0.5 | – |
| Decoder loss coeff. | $\beta_o, \beta_r, \beta_c$ | 1.0, 1.0, 1.0 | – |
| **Planner** | | | |
| Planning horizon | $H$ | 15 | – |
| Number of samples | $N_{\pi_N}$ | 500 | – |
| Mixture coefficient | $M$ | 0.05/0.0 | – |
| Number of iterations | $J$ | 6 | – |
| Initial variance | $\sigma_0$ | 1.0 | – |
| Cost budget (vision tasks) | $B$ | 2.0 | – |
| **Actor Critic** | | | |
| Sequence generation horizon | – | 15 | – |
| Discount factor | $\gamma$ | 0.997 | – |
| Reward trace | $\lambda_r$ | 0.95 | – |
| Cost trace | $\lambda_c$ | 0.95 | – |
| Learning rate | – | $3 \cdot 10^{-5}$ | – |
| Shaping coefficient | $\alpha_{\mathrm{score}}$ | 0.05 | – |
| **General** | | | |
| Number of other MLP layers | – | 5 | – |
| Units per MLP layer | – | 512 | – |
| Train ratio | – | 512 | – |
| Action repeat | – | 4 | – |

*Table 2.* Hyperparameters for SafeDreamer (Huang et al., 2024)

| Name | Symbol | Value | Description |
|---|---|---|---|
| **World Model** | | | |
| Number of latent units | – | 48 | – |
| Classes per latent | – | 48 | – |
| Batch size | $B$ | 64 | – |
| Batch length | $T$ | 16 | – |
| Learning rate | – | $10^{-4}$ | – |
| Coefficient of kl divergence in loss | $\alpha_q, \alpha_p$ | 0.1, 0.5 | – |
| Coefficient of decoder in loss | $\beta_o, \beta_r, \beta_c$ | 1.0, 1.0, 1.0 | – |
| **Planner** | | | |
| Planning horizon | $H$ | 15 | – |
| Number of samples | $N_{\pi_N}$ | 500 | – |
| Mixture coefficient | $M$ | 0.05/0.0 | $N_{\pi_\theta} = M \cdot N_{\pi_N}$, different for vector/visual input |
| Number of iterations | $J$ | 6 | – |
| Initial variance | $\sigma_0$ | 1.0 | – |
| **PID Lagrangian** | | | |
| Proportional coefficient | $K_p$ | 0.1 | – |
| Integral coefficient | $K_i$ | 0.01 | – |
| Differential coefficient | $K_d$ | 0.01 | – |
| Initial Lagrangian multiplier | $\lambda_0^p$ | 0.0 | – |
| Lagrangian upper bound | – | 0.1 | Maximum of $\lambda_p$ |
| **Augmented Lagrangian** | | | |
| Penalty term | $\nu$ | $5 \times 10^{-9}$ | – |
| Initial Penalty multiplier | $\mu_0$ | $1 \times 10^{-6}$ | – |
| Initial Lagrangian multiplier | $\lambda_0^p$ | 0.01 | – |
| **Actor Critic** | | | |
| Sequence generation horizon | – | 15 | – |
| Discount horizon | $\gamma$ | 0.997 | – |
| Reward lambda | $\lambda_r$ | 0.95 | – |
| Cost lambda | $\lambda_c$ | 0.95 | – |
| Learning rate | – | $3 \cdot 10^{-5}$ | – |
| **General** | | | |
| Number of other MLP layers | – | 5 | – |
| Number of other MLP layer units | – | 512 | – |
| Train ratio | – | 512 | – |
| Action repeat | – | 4 | – |

*Table 3.* Hyperparameters for LAMBDA (As et al., 2022). We evaluate against the official implementation; sequence generation horizon set to 15 and cost limit reduced to 2.0 for fair comparison.

| Name | Symbol | Value | Description |
|---|---|---|---|
| **Dynamics Model Ensemble** | | | |
| Sequence length | $T$ | 50 | – |
| Burn-in steps | – | 500 | – |
| Period steps | – | 200 | – |
| Number of models | – | 20 | – |
| Decay | – | 0.8 | – |
| Cyclic LR factor | – | 5.0 | – |
| Posterior samples | – | 5 | – |
| **Safety Critic** | | | |
| Learning rate | – | $2 \times 10^{-4}$ | – |
| Initial penalty | $\nu_0$ | $5 \times 10^{-9}$ | augmented penalty scale |
| Initial Lagrangian | $\lambda_0$ | $1 \times 10^{-6}$ | initial safety constraint weight |
| Penalty power factor | – | $1 \times 10^{-5}$ | – |
| Discount factor | $\gamma_c$ | 0.995 | – |
| Update steps | – | 100 | – |
| **Policy & Reward Critic** | | | |
| Learning rate | – | $8 \times 10^{-5}$ | – |
| Discount factor | $\gamma_r$ | 0.99 | – |
| TD($\lambda$) factor | $\lambda_r$ | 0.95 | – |
| Sequence generation horizon | – | 15 | – |
| **General** | | | |
| Cost limit | $d$ | 2.0 | per-episode cost constraint (adjusted from 25.0) |
| Batch size | $B$ | 32 | – |
| Action repeat | – | 4 | – |

*Table 4.* Hyperparameters for PPO-Lag (Ray et al., 2019). Hidden sizes increased to [512, 512, 512, 512] and cost limit set to 2.0 for fair comparison.

| Name | Symbol | Value | Description |
|---|---|---|---|
| **Training** | | | |
| Batch size | – | 64 | – |
| Target KL divergence | $\delta_{KL}$ | 0.02 | max KL per update |
| Max gradient norm | – | 40.0 | gradient clipping threshold |
| Critic norm coefficient | – | 0.001 | weight decay on critic |
| **Returns & Advantage** | | | |
| Discount factor | $\gamma$ | 0.99 | reward return discounting |
| Cost discount factor | $\gamma_c$ | 0.99 | cost return discounting |
| GAE lambda (reward) | $\lambda_r$ | 0.95 | reward advantage trace |
| GAE lambda (cost) | $\lambda_c$ | 0.95 | cost advantage trace |
| Advantage estimation | – | GAE | Generalized Advantage Estimation |
| **Actor-Critic Network** | | | |
| MLP hidden sizes | – | [512, 512, 512, 512] | 4 hidden layers |
| Activation | – | tanh | – |
| Learning rate | – | $3 \times 10^{-4}$ | Adam learning rate |
| **Lagrangian Safety** | | | |
| Cost limit | $d$ | 2.0 | per-episode cost constraint |
| Initial Lagrange multiplier | $\lambda_0$ | 0.001 | initial safety weight |
| Lagrange learning rate | – | 0.035 | Adam LR for $\lambda$ update |
| Lagrange optimizer | – | Adam | – |

*Table 5.* Training wall-clock time (in hours) across four Safety-Gymnasium environments. All runs conducted on identical hardware (single RTX 4090 Ti).

| Method | SafetyPointGoal2 | SafetyPointPush1 | SafetyPointButton1 | SafetyCarGoal1 |
|---|---|---|---|---|
| USB-RL (full) | 8.49 | 7.67 | 8.86 | 8.66 |
| w/o Score Distillation | 16.21 | 14.03 | 18.33 | 17.14 |
| Speedup ($\times$) | 1.91 | 1.83 | 2.07 | 1.98 |

