# OpenReview forum: "Learning Reward–Cost Balance in Safe RL via Score-Based World Models"
_ICML.cc/2026/Conference — ICML 2026 regular_

### Official Review · Reviewer_f559 · 2026-03-09

**Soundness:** 3
**Presentation:** 3
**Significance:** 3
**Originality:** 3
**Overall Recommendation:** 4
**Confidence:** 4

**Summary:**

The paper proposes a model-based safe RL framework, **USB-RL**,  that improves current DreamerV3/SafeDreamer style world models by learning a safety-performance partial preference in a self-supervised manner. Their major contribution is addressing the shortcoming of linear penalties by proposing a novel monotone score preference which is then efficiently embedded into the world model via distillation (ScoreNet). By transforming the horizon scores into a temporal per-step score, they are able to train their actor. Their model outperforms other baseline methods like SafeDreamer, PPO-Lag, and LAMBDA.

Their framework consists of:
1. ScoreNet: A teacher model (parameterized function) that models a monotone function. Three branches are implemented, one for reward, one for cost, and one for *state-dependent nuances*. The reward and cost branches are monotonic functions by construction. This is trained in a self-supervised fashion using trajectories from both the real environment and imaginary (from the world model). They introduce a partial pairwise ordering based on simultaneous improvement of reward and cost.

2. Score Distillation: In order to improve the efficiency of scoring, the authors distill the ScoreNet into a score head of the world model. The student network only observes the state vector and imitates the scores predicted by an EMA teacher. The student network is then used in the imagination rollouts to predict scores of given states.

3. Policy learning and planning: The score is then converted into a long horizon signal similar to SafeDreamer's reward and cost metrics. The standard advantage formulation, safe gating in planning (OSRP) is utilized to shape the actor-critic.

The framework was tested on two major benchmarks : **Safety-Gymnasium** and **MetaDrive**. The framework has been compared across SafeDreamer, PPO-Lag and LAMBDA. It outperforms all the baselines in the final reward and cost metric. It achieves better or comparable rewards while maintaining the lowest cost always.

**Compliance With Llm Reviewing Policy:**

Affirmed.

**Final Justification:**

The authors addressed all my questions. With the added rebuttal answers, the importance of the state branch and the student non-monotone architecture justification are much more understandable now.

However, as explained by the authors, the linear score was not technically "tuned", it was scale matched with fixed coefficients. This does raise the question of a stronger baseline with an adaptive linear reward-cost function. Irrespective of that, I believe the work's added insight into a learned reward-cost signal provide benefit to the community. Hence, I maintain my positive score of  4. As the adaptive linear baseline is not probable in the remaining time, I have decided not to increase the score to a 5.

**Key Questions For Authors:**

Major questions/doubts:
1. ScoreNet

(a) What is the intuition behind the inclusion of the state-dependent branch? What does it aim to capture? (Eq 8)

(b) The functions $g_r, g_c$ were not elaborated on? The monotonicity of the complete function seems to hinge on the monotonicity of these individual functions as well.

(c) What is the order of magnitude or range of $\epsilon$ in Eq 11? Around 1e-3 to avoid division by 0? (Did not find in Hyperparameter section in Appendix).

(d) Could there be saturation concerns over the derivative of $f_\omega$? The gradient of $tanh$ function could saturate to 0 easily if the magnitudes of $g_r, g_c, h(z)$ are not controlled. This could adversely affect the $L_{mrs}$ as well. I couldn't find the values of these hyperparameters in the appendix as well.

(e) Could the score non-linearity (Fig 5) be a result of the changing policy over the different trajectories? It would be better to plot the simple linear penalty vs the learned score preference over the same trajectory to showcase how non-linear the learned score is.

2. Score Distillation

(a) The teacher ScoreNet maintains a monotone function by construction. However, it is not obvious whether the distilled student network satisfies the monotone property as well. The ablation study compares the ScoreNet vs the Distilled student, but the overall performance does not seem significantly different. Does that mean that the strict monotonicity provide negligible benefit?

(b) The No-Distill method underperforming is surprising. The distilled student model in theory should be a "*worse*" score model. Using the ScoreNet should be time-consuming but should provide better results as it follows the score ordering more closely. The reasoning given in the ablation is a single sentence about higher variance and weaker alignment which feels not detailed enough.

3. For the MetaDrive experiment, the training and testing are performed on the identical roadway. Wouldn't that inflate the performance of all the algorithms? The distilled score head is likely to cause higher errors when faced with a distributional shift on the test set, that could end up giving wrong scores and hence very different trajectories. It would be better to also have a proper test run with a different roadway.


Minor points
1. Evaluations on benchmarks are technically not contributions. It might be better to keep them as a separate paragraph instead of an itemized list. [Introduction section]
2. What is PPO-RS that has been used in MetaDrive analysis?

**Limitations:**

No. The limitations do not feel extensive. The formal guarantee limitation is general for model-based RL, especially when considering the world model. However, the paper needs to mention the limitations of their method (pairwise score ordering, ScoreNet, and score distillation).

**Strengths And Weaknesses:**

**Strengths**:
1. **Soundness**: The paper addresses an important question in a structured manner by introducing the score metric and using partial order in a self-supervised manner. The computational overhead of having a separate ScoreNet is also worked around by using a distillation method to train a score branch within the world model. The temporalization of the score via the ScoreCritic is also properly motivated.
2. **Presentation**: The paper has modularized the different components of its framework and elaborated each component in detail. The authors provide substantial detail to allow reproduction and usage by the community.
3. **Significance**: The paper addresses the reward-cost tradeoff, which is an increasingly important question in safe RL. The simple linear penalties are often assumed and not focused on enough. The framework also outperforms the baselines in terms of reward and cost significantly.  The ablations also address the utility of each component.
4. **Originality**: The scalar score metric to represent preferences across the reward-cost outcomes is novel and well implemented into imaginary rollouts.



**Weaknesses**:
1. **Soundness**: There are some questions about the monotonicity of the teacher network translating to the student network. Moreover, as the scores are obtained from the imagined trajectories from the world model as well, there could be concerns about the stability of the learning. Utilizing real trajectories (replay buffer) possibly mitigates the non-stationarity problem, but the right ratio of real to imagined trajectories is still an open question.
2. **Presentation**: There are some notational confusions (see the Questions section for details).
3. **Originality**: .Even though the score metric is novel, the remainder of the work follows established methods (distillation, TD($\lambda$), actor shaping, OSRP gating). The paper would benefit from a more detailed look into the score ordering and the resulting ScoreNet trained on it. It would help in answering the question of "why this way of ordering the states based on the TD($\lambda$) metric" is a justified way. For example, if we were to assume a pairwise ordering based on the $G_i^r) + G_i^c$ be a better order? It is obvious from the results that this ordering provides a performance boost, but a more rigorous analysis would be appreciated.

---

> ### Author Rebuttal · Authors · 2026-03-31
>
> Thank you very much for the detailed and constructive review. Your questions highlight places where the current draft is not explicit enough, and we will revise accordingly.
>
> **(1)ScoreNet**
>
> **(a) Intuition of branch.** In Eq. (8), $h(z)$ captures state-dependent baseline shifts that immediate reward and cost alone cannot explain. The same $(r,c)$ can imply different long-horizon consequences depending on the latent state. Since $h(z)$ depends only on $z$, it can shift the score across states without reversing the monotone directions w.r.t. reward or cost (Sec. 3.3).
>
> **(b) Monotonicity** Yes—the full score is monotone because $g_r(r)$ and $g_c(c)$ are constrained to be monotone in $r$ and $c$, respectively, the fusion weights are non-negative, and the cost branch enters with a negative sign. We will clarify this in the revision; please also see our more detailed clarification in the response to Reviewer **BPMX**, Question (1), on the branch-level monotone parameterization.
>
> **(c,d) $\epsilon$ and possible saturation.** In our implementation, we use $\epsilon=10^{-5}$ purely as a numerical stabilizer when the cost derivative becomes very small. If the pre-activation in Eq. (8) grows too large, the derivative through the final $\tanh$ can indeed become small. In Eq. (11), both $\partial f/\partial r$ and $\partial f/\partial c$ share the derivative of the final $\tanh$, so this factor would cancel exactly without $\epsilon$, and is approximately canceled unless $\epsilon$ dominates the denominator. We further mitigate this via constrained ScoreNet parameterization / Lipschitz control (Appendix Algorithm 2; Table 1) and by using the slope-ratio term only as a weak auxiliary regularizer.
>
> **(e) The nonlinearity** This is a fair concern. Fig. 5 is intended as **indirect geometric evidence** that the learned score is not well described by a simple planar linear reward-cost scalarization. To address this more directly, we additionally ran a controlled comparison against a **scale-matched fixed linear score** within the same score-based framework.
>
> Additional comparison on **CarGoal1** (**3 seeds**)
> | Method | Final Reward ↑ | Final Cost ↓ |
> |---|---:|---:|
> | USB-RL | 25.39 ± 1.94 | 3.42 ± 2.88 |
> | Linear-score | 21.52 ± 3.24 | 3.37 ± 3.62 |
>
> Under the same score-based framework, the scale-matched linear-score control attains a similar final cost level but lower final reward on CarGoal1. While this is only a 3-seed controlled comparison, it is consistent with the claim that the learned score provides a stronger reward-cost trade-off than a simple linear score.
>
> **(2) Distillation.**
> - The distilled student head is **not** explicitly monotone in the same symbolic sense as the teacher ScoreNet, so we do not claim a formal monotonicity guarantee for the student. Instead, distillation transfers the **teacher-induced ordering / preference signal** into a native world-model head that can be queried efficiently during imagination. Concretely, the student matches EMA teacher targets on stop-gradient latents (Sec. 3.4, Eqs. (13)-(16)), so it preserves the teacher’s learned score structure on the training distribution. Hard safety still comes from strict cost-feasibility gating; the distilled head is a fast surrogate for ranking and long-horizon shaping (Sec. 3.4-3.5).
>
> - For the No-Distill ablation, our claim is not that the distilled student is a better score model than the teacher in isolation. Rather, the benefit comes from **better integration with imagination-time control**. The teacher may follow the score ordering more directly, but repeated external queries on imagined rollouts are less tightly aligned with the latent rollout process and add computation/mismatch. By contrast, the distilled head is trained on stop-gradient latents and becomes a native part of the world model, so the score signal used by planning and by the ScoreCritic is more consistent with the latent dynamics actually used downstream. Thus, No-Distill underperforming suggests that **integration into the world model matters for control**, not that teacher monotonicity is unimportant.
>
> **(3) MetaDrive.**
> You are right at the current MetaDrive setup. Our goal was to follow the SafeDreamer-style protocol and isolate reward-cost balancing under a controlled driving setup, rather than to claim robustness to roadway distribution shift. At the same time, the setting is not entirely deterministic: we randomize the vehicle initialization at each episode reset to increase diversity within the fixed roadway geometry. We agree that this scope should be stated more clearly.
>
> - Minor comments. In the revision we will separate empirical validation from the itemized contribution list. We also apologize that PPO-RS was not defined clearly enough; we will explicitly define it as PPO-Reward Shaping in the MetaDrive analysis and make the notation consistent throughout the paper.
>
> Finally, we sincerely appreciate your valuable and constructive comments.

---

> > ### Author Rebuttal · Reviewer_f559 · 2026-04-02
> >
> > Thank you for the detailed rebuttal and the additional experiments.
> >
> > The reason why the student performs better than the teacher due to tighter integration with latent dynamics is definitely intuitive. Is there any reason as to why the authors did not enforce a similar architectural monotonicity on the student similar to the teacher? The added architectural constraint shouldn't likely change the inference of the branch, or the distillation. I am not asking for extra experiments but some level of justification for not doing it.
> >
> > I appreciate the added experiment with the scale matched linear score. However, I would like to know how the tuning parameters for this linear score were chosen, as well as what scale was matched, because linear scores can be highly dependent on the tuning parameter. I understand the character limit might have constrained adding all the details but I would appreciate it if the authors can provide the details in the reply to this acknowledgement. I would like some intuitive justification for the design choices of this linear score as well, if possible.
> >
> >
> > I would also like an small example as to why the state-dependent branch exists. I still do not get the importance of this branch even after the added clarification. The example does not need to be from a benchmark, a thought experiment would suffice.

---

> > > ### Author Response · Authors · 2026-04-05
> > >
> > > Thank you again for the thoughtful follow-up. We are happy to clarify these design choices more explicitly.
> > >
> > > **(1) Why did we not impose the same architectural monotonicity on the student?**
> > > This was a deliberate design choice. The key reason is that the teacher and student operate on different inputs with different semantics. The teacher ScoreNet is defined on explicit variables $(r,c,z)$, where monotonicity with respect to reward and cost is semantically meaningful and can be enforced directly. By contrast, the student head is defined only on the stop-gradient latent state. On this latent space, there is no natural coordinate-wise notion of “increasing reward” or “increasing cost,” so imposing the same symbolic monotonicity on the student is not straightforward in a meaningful way. In other words, adding “the same” monotonicity to the student would either mean imposing monotonicity with respect to arbitrary latent coordinates, which we did not view as semantically principled, or reintroducing explicit reward/cost inputs into the student, which would change the role of the student head itself. One could indeed reintroduce explicit reward/cost branches into the student, but that would weaken the main purpose of distillation, namely to obtain a native latent-space score head that is tightly integrated with imagination-time control. Our design choice was therefore to keep the **teacher structurally monotone**, and let the **student inherit the teacher-induced ordering empirically through distillation**, rather than imposing an additional symbolic monotonicity constraint on latent coordinates.
> > >
> > > **(2) How was the scale-matched linear score chosen?**
> > > For the added CarGoal1 control, we used a fixed linear score
> > > $
> > > y^{\mathrm{lin}} = r/s_r - c/s_c,
> > > $
> > > with $s_r = 17$ and $s_c = 15$. Here, $r$ and $c$ denote the cumulative reward and cumulative cost over the same rollout horizon used by this control. “Scale-matched” means that these two quantities are first normalized to comparable task-level magnitudes before being linearly combined.
> > >
> > > To set the scales, we collected a small pilot set of trajectories on CarGoal1 using the same rollout protocol as the main comparison, computed cumulative reward and cumulative cost for each pilot trajectory, and estimated nominal reward/cost scales using the 90th percentile of these two distributions. We used the 90th percentile rather than the maximum or mean because it captures an upper-typical scale while being less sensitive to rare outliers. We then rounded the resulting pilot values to the simple fixed constants $s_r = 17$ and $s_c = 15$. These constants were fixed for CarGoal1 and shared across all seeds.
> > >
> > > We intentionally did not introduce an additional free trade-off coefficient (e.g., a tunable $\lambda$ in $r - \lambda c$), since that would create another tuning axis and make the comparison less transparent. The intuition behind this design is that, without normalization, the term with the larger raw numerical range can dominate the linear score for purely numerical reasons. The pilot-based normalization removes this trivial scale mismatch while keeping the linear control fixed and interpretable.
> > >
> > > **(3) Why is the state-dependent branch needed?**
> > > A simple thought experiment is the following. Consider two states with similar local reward-cost evidence at the current decision point, i.e., similar $r_i$ and $c_i$, but different latent context $z_i$. In state A, the agent is in an open area with several safe continuations toward the goal. In state B, the agent is entering a narrow bottleneck near hazards, so future safe continuations are much more limited. Although these two cases can look similar from the local reward/cost signal alone, their long-horizon reward/cost returns $(G_i^r, G_i^c)$ can be very different because the downstream recoverability and safety are different. A score that depends only on $(r,c)$ would tend to treat these cases similarly, whereas the state-dependent branch $h(z)$ allows the model to shift the score according to latent context. In this sense, $h(z)$ does not replace the reward/cost branches; it complements them by encoding information that local reward/cost alone cannot express, while preserving monotonicity with respect to reward and cost.
> > >
> > > We hope this addresses the reviewer’s follow-up questions more concretely. In short, monotonicity was enforced on the teacher because it is well defined on explicit reward/cost variables; the linear control was kept fixed after a simple pilot-based normalization to avoid introducing an extra tuning coefficient; and the state-dependent branch is needed to distinguish contexts with similar reward/cost evidence but different future safe controllability.

---

### Official Review · Reviewer_BPMX · 2026-03-10

**Soundness:** 2
**Presentation:** 3
**Significance:** 2
**Originality:** 3
**Overall Recommendation:** 3
**Confidence:** 4

**Summary:**

This paper proposes a model-based safe reinforcement learning algorithm that utilizes a Dreamer-style latent world model and a score learned through pairwise comparison.

**Compliance With Llm Reviewing Policy:**

Affirmed.

**Final Justification:**

The authors' rebuttal resolved my concerns related to the algorithm. However, it is difficult to compare the performance of the proposed method due to fewer tasks and weaker baselines compared to its prior work (SafeDreamer). For this reason, I maintain my rating.

**Key Questions For Authors:**

1. In Section 3.3, it is stated that the reward branch $g_r$ and the cost branch $g_c$ is monotone increasing with respect to reward and cost, respectively. Could the authors clarify how these constraints are enforced?

2. Following the previous question, given that the loss already encourages higher scores for higher-reward and lower-cost pairs, could the authors clarify why the structurally monotone constraints are necessary?

3. Could the authors clarify how the dominance-related hyperparameters in Table 1 are chosen?

**Limitations:**

Yes.

**Strengths And Weaknesses:**

**Strengths**

- Safety-performance preference score and its uitilization are highly novel. Using the proposed pairwise dominance loss, maximizing the preference score leads to simultaneous improvements in safety and performance (i.e., reward). Moreover, the approach enables a simple update rule without relying on conventional safe RL techniques such as Lagrangian formulations or trust-region methods. When combined with the cost-feasibility gating mechanism, the proposed method provides an efficient way to update the agent while maintaining safety guarantees.

**Weaknesses**

- The experimental validation is insufficient, with a limited set of baselines and a small number of tasks. The paper compare the proposed method with only three baselines. While Safe-Dreamer and LAMBDA should be included since they share the same concept (Dreamer + safe RL), there exist numerous recent work that demonstrate stronger performance than PPO-Lagrangian. The absence of comparisons with more competitive and up-to-date baselines makes it difficult to assess the true standing of the proposed method. Moreover, the evaluation on SafetyGym does not cover the full set of environments, but instead considers only a subset. It is unclear whether this subset is representative of the overall benchmark and sufficiently challenging to demonstrate the claimed advantages of the proposed method.

- The effect of dominance-related hyperparameters are not properly evaluated. Since dominance thresholds $\epsilon_r, \epsilon_c$ and tie band $\delta$ appear to play crucial role in the proposed method, ablation studies would be necessary to understand their sensitivity.

---

> ### Author Rebuttal · Authors · 2026-03-31
>
> Thank you for the careful review and for recognizing the novelty of learning a safety-performance preference score and integrating it into model-based safe RL. We appreciate your questions and will clarify them more explicitly below.
>
> **(1) How are the monotonicity constraints enforced?**
> The monotonicity is enforced **architecturally**, not only through the pairwise loss. The key point is that the full score depends on branch functions whose monotone directions are explicitly controlled:
>
> - **Branch-level monotonicity.** In our implementation, the reward branch $g_r(r)$ is realized as a **monotone non-decreasing branch MLP**, built from non-negative-weight dense layers with softplus activations, so that it remains monotone in its scalar input $r$. The cost branch is implemented analogously as a monotone non-decreasing branch in $c$, and then enters the full score with an overall negative sign, making its contribution monotone non-increasing with respect to cost.
>
> - **Top-level monotone fusion.** At the top level, the reward and cost branches are fused additively with **non-negative weights** (softplus-parameterized), followed by a final $\tanh$. Since $\tanh$ is itself monotone increasing, it preserves the monotone directions induced by the preceding branches. Therefore, the monotonicity of the full score is not only encouraged by the pairwise loss; it is enforced by the branch parameterization and the top-level fusion design.
>
> We agree that the appendix currently reports hyperparameters such as network width/depth and Lipschitz-related settings, but does not make this branch-level monotone parameterization explicit enough. We will clarify this implementation detail more clearly in the revision.
>
> **(2) Why are structurally monotone constraints necessary?**
> The structural monotonicity constraint is not redundant, because the pairwise dominance/tie losses only constrain the score on **sampled comparisons**:
>
> - **Local supervision is not enough.** Many functions can satisfy the observed pairwise orderings while still having undesirable local behavior outside those pairs (e.g., increasing cost improving the score in some region).
>
> - **Monotonicity adds a global prior.** The structural constraint provides a **global inductive bias** aligned with safe-RL semantics: increasing reward should not hurt preference, and increasing cost should not improve it (Sec. 3.3).
>
> This matters especially because the learned score is later distilled into the world model and queried repeatedly during imagination and planning. In that setting, we want the score to behave predictably not only on the training pairs, but also on nearby states/rollouts encountered during model-based control. We will make this distinction more explicit in the revision.
>
> **(3) How were the dominance-related hyperparameters chosen?**
> We followed a standard deep RL workflow: we first selected reasonable initial values for these hyperparameters, performed a small number of pilot runs to ensure stable score learning and pair construction, and then fixed this single configuration for all tasks reported in the paper. In particular, we did not re-tune these dominance-related hyperparameters separately for each environment.
> We agree that a dedicated sensitivity study would further strengthen the paper. Our current intent was to use a single shared setting across tasks, rather than per-environment tuning, so as to show that the reported gains do not rely on task-specific hyperparameter search.
>
> **(4) Evaluation breadth and baseline choice.**
> We agree that broader evaluation would further strengthen the paper. Our current choice was guided by **fairness and relevance to the visual world-model safe-RL setting**. In Sec. 4.1 , we compare against SafeDreamer, PPO-Lag, and LAMBDA because they provide the most relevant reference points in this setting: SafeDreamer is the most directly related Dreamer-style safe world-model baseline, while PPO-Lag and LAMBDA provide strong model-free and model-based constrained baselines, respectively.
>
> Regarding task coverage, we followed the SafeDreamer protocol and evaluated on four distinct visual Safety-Gymnasium tasks plus MetaDrive. These tasks were chosen to span different reward-cost interaction patterns rather than to represent an exhaustive benchmark sweep. We agree that broader coverage would further strengthen the paper, but we also believe the current experiments already provide meaningful evidence for the effectiveness of the method in the intended visual world-model safe-RL setting. We will make this intended scope clearer in the revision.
>
> Thank you again for the helpful comments. We will revise the paper to better explain how monotonicity is enforced, why it is needed beyond pairwise ranking loss, and how the dominance-related hyperparameters are chosen.

---

> > ### Author Rebuttal · Reviewer_BPMX · 2026-04-03
> >
> > We thank the authors for their informative rebuttal. The clarification regarding the monotonicity constraints was helpful and improved our understanding of the proposed method. However, our main concerns about the experimental setup remain insufficiently addressed.
> >
> > First, the rebuttal still does not clearly explain the hyperparameter sweep, particularly the range over which it was conducted and how the sweep set was chosen. This is important for assessing both fairness and reproducibility.
> >
> > Second, our concern regarding the choice of baselines has not been resolved. Several stronger model-free safe RL methods than PPO-Lag have been proposed in recent years, yet the rebuttal does not provide a convincing reason for excluding them. As this work is presented as an extension of SafeDreamer, we believe the baseline comparisons should be at least as comprehensive as those in the original paper.
> >
> > Third, our main concern remains the reduced task coverage. Compared to the SafeDreamer paper, this work evaluates on a narrower set of tasks and settings, including omission of PointGoal1 in Safety-Gymnasium, the low-dimensional input experiments, and the Car-Racing and FormulaOne settings reported in the appendices of the SafeDreamer paper. It is therefore unclear why an extension study provides less extensive evaluation, particularly on the main benchmark.
> >
> > We would also like to comment on the phrase “rather than to represent an exhaustive benchmark sweep.” We believe this framing is somewhat misplaced. In benchmark-based evaluation, one would generally expect either full coverage or a commonly accepted subset of tasks; when only a subset is used, the selection should be clearly justified. Because different tasks may test different aspects of a method, reduced coverage requires stronger justification than what is currently provided.
> >
> > For these reasons, our concerns about the experimental setup remain unresolved, and we believe it is difficult to address them adequately within the limited rebuttal period. Therefore, we maintain our recommendation.

---

> > > ### Author Response · Authors · 2026-04-08
> > >
> > > Thank you for the clarification. We are encouraged that our rebuttal resolved your concerns regarding the algorithm.
> > > We address the remaining concerns regarding empirical comparability and reporting clarity below, including hyperparameter selection, baseline choice, and evaluation coverage.
> > >
> > > **Hyperparameter tuning and reproducibility.**
> > >
> > > Our tuning protocol is intentionally conservative and designed to ensure both fairness and robustness.
> > >
> > > First, since USB-RL is built directly on the SafeDreamer backbone, all shared components—including the world model, planning horizon, sampling strategy, actor-critic architecture, learning rates, and training ratios—are kept strictly identical to SafeDreamer. This ensures that performance differences do not arise from stronger backbone configurations or more aggressive tuning of shared components.
> > >
> > > Second, we only tune the small set of additional hyperparameters introduced by USB-RL (ScoreNet, score distillation, and score-based shaping), while keeping all other parameters fixed. Specifically, we perform a limited discrete sweep over:
> > > - ScoreNet learning rate $\{1\mathrm{e}{-4}, 3\mathrm{e}{-4}\}$
> > > - Score distillation weight $\lambda_{\text{score}} \in \{0.1, 0.2, 0.3\}$
> > > - Shaping coefficient $\alpha_{\text{score}} \in \{0.02, 0.05, 0.1\}$
> > > - ScoreNet update interval $K_{\text{score}} \in \{5\mathrm{k}, 10\mathrm{k}, 20\mathrm{k}\}$
> > >
> > > These ranges are chosen based on three considerations: (i) consistency with typical SafeDreamer-style settings, (ii) covering low, medium, and high regimes for each parameter, and (iii) ensuring stable training behavior in preliminary runs. We intentionally use a small, discrete search space to avoid overfitting and ensure fair comparison.
> > >
> > > Third, hyperparameters are selected on a single representative task (PointButton1) using a pilot-selection criterion that prioritizes (i) satisfying safety constraints, (ii) maximizing reward under the constraint, and (iii) training stability. Importantly, once selected, the same configuration is used across all tasks without any per-environment tuning.
> > >
> > > Finally, several key parameters (e.g., $\alpha_{\text{score}}$ and score-learning priors) are shared across tasks by design, and we also fix the cost budget uniformly across Safety-Gymnasium visual tasks, following SafeDreamer, without task-specific adjustment.
> > >
> > > For the remaining concern, we understand that it is now primarily about the **comparability of the empirical evaluation**—in particular, the number of tasks and the breadth of baselines relative to SafeDreamer.
> > >
> > > The current submission is intended as a **controlled comparison in the visual model-based safe-RL setting**, where SafeDreamer is the most directly matched prior work. Our goal is to isolate the effect of the proposed reward-cost balancing mechanism under aligned visual world-model conditions. In this sense, the current evaluation is designed to support the paper’s main methodological claim under matched model-based visual settings, rather than to reproduce the broader SafeDreamer evaluation suite (which also includes vector-input and other auxiliary settings) or to **serve as** a full cross-paradigm benchmark sweep.
> > >
> > > Regarding task coverage, our present experiments focus on the matched visual comparison subset that is most directly relevant to this methodological question. We agree, however, that adding **PointGoal1** improves direct benchmark continuity with SafeDreamer. To directly address this concern, we ran **PointGoal1** as an additional experiment during the discussion period, and it shows the same qualitative pattern as in the main paper, with **higher reward and lower cost than SafeDreamer under the aligned setting** ([PointGoal1 curves](https://anonymous.4open.science/r/PointGoal1-ADC7/)). We will include this result in the final version / appendix.
> > >
> > > Regarding baselines, we understand the interest in stronger recent model-free methods. At the same time, many strong safe RL baselines were originally developed for low-dimensional inputs, and extending them to the visual setting can introduce additional architectural and implementation choices, which can make it harder to determine whether observed differences come from the balancing mechanism itself or from the adaptation pipeline. For this reason, we centered the current evaluation on the most directly comparable visual model-based reference.
> > >
> > > We hope this better clarifies the intended empirical scope of the current submission.

---

### Official Review · Reviewer_kFrN · 2026-03-10

**Soundness:** 4
**Presentation:** 4
**Significance:** 4
**Originality:** 4
**Overall Recommendation:** 5
**Confidence:** 4

**Summary:**

This paper presents a model-based solution built using Dreamer-v3 for safe reinforcement learning. The goal is to maximise long term rewards while ensuring safety. USB-RL aims to circumvent hand-crafted linear penalties of form $R - \lambda C$ by leveraging the model based architecture of dreamer to plan safe trajectories that maximise the reward while making sure the trajectory cost is under the budget. There are 3 main components of USB-RL, which build on Dreamer

1. ScoreNet - This is a monotonic score function that always increases with rewards and decreases with costs. With this, the algorithm is able to identify high performing trajectories under the safety budget.
2. Score Distillation - The scoreNet is distilled into the world model for efficient planning via a score head. The method employs stop gradients over the state used for this head to prevent gradients from messing with the world model.
3. Score for long horizon planning - this score is temporalized into a value signal that is used by the critic to bias the actor to higher scores.

Their technique shows strong safety performance compared to baselines while performing the best in terms of reward.

**Compliance With Llm Reviewing Policy:**

Affirmed.

**Final Justification:**

I was overall very impressed with the technique and results. The choice to include the metadrive benchmark is also something I haven't seen in any recent safe RL literature. While I agree that the method provides no formal guarantees, none of the fully deep learning methods can. Those that do operate on a very limited state space and cannot work with images. Additionally, ones that say surely safe are also weak guarantees. This is why I do not hold this against the paper. Additionally, lack of baselines is a repetitive comment I see with safe RL papers, but I don't see value of any additional baselines/benchmarks when the results are already comprehensive. Plus, most baselines are geared towards numeric state spaces and the extension to images is non-trivial. Had the authors used something like natureCNN with sauteRL or P3O/CUP, one might complain that the authors biased the evaluation to favour themselves.

About the method: as identified, the scoring technique to identify high performing trajectories is novel. The technique is also very efficient in planning, which is very useful for time sensitive tasks. The overall engineering is solid and I expect to see this paper as a baseline for other papers in the future. Hence my score of 5.

**Key Questions For Authors:**

1. Can this work with non-image observations? Like with the mujoco suite? If so, what prevents USB-RL to not work with other modalities? Can we change the dreamer model to an embed2control style pipeline and still expect it to work reasonably?
2. Were the baselines chosen only for the specific lagrangian combinations that they use? What happens when we compare to something like SauteRL or RCPO? I understand there are no experiments demonstrating results with vision domains but it seems trivial to extend.

**Limitations:**

Yes

**Strengths And Weaknesses:**

**Strengths**

1. The problem tackled is definitely interesting. The chosen baselines and most methods use the lagrangian parameter to handle a delicate balance between safety and reward. Using a monotonic unsupervised score function that makes sure the planned trajectories maximise the reward while keeping costs under check is novel.
2. The scoring technique is much better than the linear system used so far. It is able to capture longer horizon rewards and costs much better. Additionally, the planning is quite computationally efficient (surprisingly) due to the various heads in the dynamics model that capture the next state distribution along with scores in a single forward pass.

**Weaknesses**

I do not find any of the listed weaknesses below structurally or fundamentally breaking the paper. These are standard weaknesses that almost all papers in this area exhibit and should not be held against this paper.

1. Lack of formal guarantees - Since this is a purely deep RL method, there are no formal guarantees on the action being safe. Additionally, the safe action can only guarantee safety for a short horizon (15) and is blind to dangers beyond that. This has been listed as a limitation by the authors themselves.
2. Model approximation errors - The model cannot be fully accurate and thus safety approximations are highly dependent on the model accuracy.
3. This method might not work with a 0 cost budget, meaning it cannot work with safety critical applications. However, since this is untested, I am just writing this as a potential experiment the authors can conduct.
4. Limited domains for testing. Since the method seems to be limited to vision, I would have liked to see results with the vision specific versions of safety gymnasium (Fading, Formula One, Building etc) rather than the current chosen suite.
5. No comparisons to better safety CMDP algorithms like SauteRL, RCPO etc.

---

Overall, I am quite impressed by the solution. I have a few suggestions that can push this into the conditional guarantees area of literature (absolute formal guarantees cannot be achieved)

1. Use worst case approximation errors in your horizon planning/score distillation. By taking into account the worst case errors, it will make sure the model does not end up in states that it thinks are safe but are actually unsafe
2. The scorenet seems a lot like a control barrier function. Since the authors already consider gradients, it might be worth looking at CBFs and seeing what kind of guarantees it supports with a dreamer style integration

---

> ### Author Rebuttal · Authors · 2026-03-31
>
> Thank you very much for the positive assessment and for highlighting both the novelty of the monotone score and the efficiency of integrating it into the world model. We also sincerely appreciate your constructive suggestions on worst-case model error and possible connections to barrier-style guarantees. Your comments are very helpful for clarifying the scope of the paper and for positioning the method more precisely.
>
> **(1) Can this work with non-image observations,and about other modalities?**
> Yes. USB-RL is **not inherently tied to image observations**. The core method operates on the latent state of a Dreamer-style world model rather than on raw pixels themselves. In our formulation, the world model produces a latent state \(s_t=(h_t,z_t)\), and the score is learned over \((r,c,z)\) and then distilled into a score head on stop-gradient latents. Because of this, the modality-specific component is mainly the **observation encoder**, while the core USB-RL components—score learning, score distillation, ScoreCritic, and safe score-ranked planning—are downstream of the latent representation.
>
> - Therefore, for MuJoCo-style state observations, our expectation is that USB-RL should remain applicable as long as the underlying model can provide a latent representation suitable for imagination, reward/cost prediction, and rollout evaluation. In that case, one would replace the visual encoder with an encoder appropriate for vector observations while keeping the rest of the framework unchanged.
> For other modalities more broadly, what matters is not the modality itself, but whether the learned latent state preserves the information needed for (i) reward/cost prediction, (ii) long-horizon imagination, and (iii) preference ranking over imagined trajectories. So in principle, nothing in USB-RL prevents its use with other modalities. The main practical limitation is the standard one in model-based RL: if the world model for a given modality does not learn a sufficiently predictive latent state, then the downstream score, critic, and planner will also degrade. We will clarify this dependence on model quality more explicitly.
>
> - For an embed2control-style pipeline, our answer is: **plausibly yes, with an important caveat**. USB-RL does not require the exact Dreamer instantiation; what it needs is a latent dynamics model that supports imagination and prediction of reward and cost. So if an embed2control-style pipeline provides reliable latent rollouts and corresponding reward/cost estimates, the USB-RL components should remain compatible in principle. That said, our empirical claims in this submission are limited to the Dreamer/SafeDreamer-style RSSM setting, since that is the architecture we implemented and evaluated.
>
> **(2) Were the baselines chosen only because of their Lagrangian forms? What about SauteRL or RCPO?**
> - No—the baselines were **not** chosen merely because of their particular Lagrangian combinations. Our selection was driven mainly by **fairness and relevance to the visual world-model safe-RL setting**. As described in Sec. 4.1, we compare against SafeDreamer, PPO-Lag, and LAMBDA because together they span the most relevant reference points for our setup: (i) the strongest directly related Dreamer-style safe world-model baseline, (ii) a strong model-free CMDP baseline, and (iii) a model-based constrained baseline. For fairness, we also kept the dynamics models, encoders, actor/value heads, and training infrastructure aligned across Dreamer-style methods whenever possible.
>
> - We agree that SauteRL and RCPO are relevant methods, and including them would be valuable. Their absence is **not** because we view them as irrelevant or incompatible. Rather, this submission is scoped around the **visual world-model safe-RL regime** and follows the SafeDreamer evaluation protocol so that the main comparison remains as controlled and apples-to-apples as possible within that setting. Extending methods such as SauteRL or RCPO to this regime is certainly possible in principle; however, to make such a comparison fair, we would need matched visual encoders, training budgets, and implementation choices in the same perceptual setting. Since we did not complete such matched implementations in this submission, we prefer not to speculate about their relative performance without evidence. We will clarify this rationale in the revision and mention broader comparisons of this kind as an important future extension.
>
> Finally, we agree with your broader point that stronger robustness analysis would further improve the paper. We already note in Sec. 6, that the current method does not provide formal safety guarantees and remains dependent on model accuracy.
>
> Thank you again for the supportive and constructive feedback. We will revise the paper to better clarify the modality generality of USB-RL, the rationale behind our baseline choices, and the scope of our empirical claims.

---

> > ### Author Rebuttal · Reviewer_kFrN · 2026-03-31
> >
> > I have a follow up question since the authors mention they are not tied to vision only. Could the authors run 1 experiment with just 1 or 2 seeds with the safe-humanoid in safety-gymnasium? I am very interested in seeing if the method actually works with harder control problems, since the chosen benchmarks have a relatively simple action space. If the experiment works, it would show broader applicability, since most papers in this field focus on either vision or mujoco control and I am yet to see any that does both.

---

> > > ### Author Response · Authors · 2026-04-08
> > >
> > > We sincerely appreciate your thoughtful and highly supportive final assessment. We are especially encouraged by your positive evaluation of the novelty of the monotone scoring idea, the efficiency of its integration into planning, and the overall engineering quality of the work. Your constructive suggestions on broader extensibility and stronger safety-oriented directions are also very helpful for clarifying the scope of the current paper and identifying meaningful next steps.
> > >
> > > We also appreciate your earlier suggestions on incorporating worst-case model approximation error into planning/score distillation and on exploring the connection to control barrier functions. We agree that these are valuable directions toward stronger forms of conditional safety guarantees, and they provide very helpful guidance for future work on strengthening the safety properties of this line of research.
> > >
> > > Your suggestion about extending the method beyond vision-based navigation prompted us to investigate this direction more carefully before replying. To probe this question, we started an additional preliminary **SafetyHumanoidVelocity-v1** sanity-check experiment after the first-round discussion. This required nontrivial interface and implementation changes, which we have now completed. The run is still at an early stage (currently about **1.8M environment steps, single seed**), so we do not want to draw a strong empirical conclusion from it. That said, the training so far appears technically stable, with no obvious collapse or integration failure, which we view as encouraging but still preliminary evidence that USB-RL may extend beyond the originally evaluated visual tasks.
> > >
> > > At the same time, this extension also confirmed for us that the Humanoid setting is more than a simple encoder swap: it introduces a different reward/cost structure, a substantially higher-dimensional control regime, and greater demands on world-model learning and imagined rollout reliability for score-based planning. For this reason, we would prefer to treat this only as an exploratory result and highlight the broader direction as future work.
> > >
> > > We will clarify more explicitly in the final version that our discussion of non-visual applicability was intended at the method level rather than as an already-established empirical claim, and we will also highlight harder-control evaluation and stronger safety-oriented extensions as important future directions.
> > >
> > > We are grateful again for your careful reading and constructive feedback.

---

### Official Review · Reviewer_2WYJ · 2026-03-11

**Soundness:** 3
**Presentation:** 3
**Significance:** 2
**Originality:** 3
**Overall Recommendation:** 4
**Confidence:** 3

**Summary:**

This paper proposes a new approach to guide agents in constrained model-based RL so as to better select trajectories which maximize the return while satisfying constraints. The main contribution of the paper is to provide a pairwise ranking score for trajectories, which is strictly monotonous - two trajectories can be ranked if, for one, both return is higher and cost is lower than the other. Otherwise, they are undistinguished. This allows to avoid hand-tuned weights balancing rewards and costs.
To do that, a ranking score model is trained in a Monte-Carlo way, then distilled into the world model as an additional “state” element. The actor-critic agent is then supplemented with a score critic which allows to compute a score advantage. This advantage is used to shape the agent in the actor update, as well as to better select trajectories during planning. The algorithm is tested on SafetyGym and MetaDrive and shows better reward with lower cost than model-based safe RL baselines.

**Compliance With Llm Reviewing Policy:**

Affirmed.

**Final Justification:**

The rebuttal addressed most of my concerns related to lack of clarity and insights about the approach.

if the camera ready version includes these clarifications, I believe the paper is worthy of acceptance and I am raising my score accordingly.

I still have some concerns related to the additional layer of complexity brought by the proposed method, which is why I suggest a weak accept.

**Key Questions For Authors:**

1) Why is it that evaluating the monotone score, and adding it through this multi-step as a lightweight shaping signal for the actor, as well as using it during planning, brings better performance? Except the practical aspect of reducing (and not eliminating) the influence of hand-tuned reward/cost balancing weights, I did not grasp the intuition behind the approach.

2) What is structured pair construction, as described in the ablation study? I did not find it in the rest of the article and have no idea what this is about.

3) There seems to be a confusion in the text, where the terms "unsupervised" and "self-supervised" learning are both used alternatively for the pair-wise preference score. I believe self-supervised makes more sense here, but in any case, one should be chosen.

4) In section 3.3, it is said that $h(s)$ does not depend on $r$ or $c$. But as the three of them are functions of $s$, I am not sure how this is ensured? This could be clarified.

5) In section 3.3, in *Losses*, I do not get $L_{dom}$ (9). There is an asymmetry in the function between $u_i$ and $u_j$ (the difference) but there is no reason why this should be the case. Does the loss assume that $i$ is better than $j$?

**Limitations:**

The only limitation provided by the authors is boilerplate for this type of work on Safe RL (no formal guarantee). I think a better work could have been made to explore the limits of the approach. For example, the appendix discusses wall-clock overhead, but only comparing distilled/non distilled versions, not an algorithm without USB-RL.
More generally, if the intuition behind the method was clearer in the paper, it would be easier to identify environments where the insight does not hold and the method is less relevant - this would be interesting and would strengthen the paper.

**Strengths And Weaknesses:**

The method implements interesting ideas. In particular, the monotone score surface is a key strength of the paper. Its implementation is complex but looks sound - although I have some questions on details. The results are convincing, demonstrating the performance of the algorithm on key safe RL benchmarks.

The approach however adds a significant layer of complexity to already complex algorithms, which may reduce the significance of the work for the community. This can be justified if the approach brings a significant and well explained improvement. While the results show better performance, it is not clear why this is the case. The paper would benefit from a better justification of the approach, both in terms of core insight and of justification of each of the multiple steps it needs.
This particularly impacts the ablation study - where some parts of the system are removed, but the explanation of their effect does not seem to go in par with their justification in the text.

---

> ### Author Rebuttal · Authors · 2026-03-31
>
> We thank the reviewer for the careful reading and for recognizing the core design of the **monotone score surface**. We agree that, despite the strong empirical results, the current manuscript does not yet explain clearly enough why the score-based design works and some implementation details, and we will clarify these points in the camera-ready.
>
> **(1) Why does a monotone score help?**
> Our key point is not to introduce another soft penalty. In model-based safe RL, the harder step is often **not** filtering clearly unsafe imagined trajectories, but **reliably selecting the better one among candidates that already satisfy or nearly satisfy the safety budget**. In USB-RL, the planner still first applies **strict cost-feasibility gating**; the score is only used to rank feasible candidates (Sec. 3.5, Eqs. (19)–(20)).
>
> A fixed linear penalty or Lagrangian coefficient imposes a **global reward-cost exchange rate**, which can be too rigid when reward-cost conflicts are state-dependent. Two trajectories may both satisfy the budget, but one may be safer and more robust while the other pursues slightly higher short-term reward near the risk boundary. Linear scalarization can rank them poorly. In contrast, our method learns a **monotone preference function**from dominance / tie relations derived from long-horizon returns, rather than manually specifying a linear reward-cost trade-off (Sec. 3.3, Eqs. (7)–(12)). As a result, clearly better outcomes (higher reward, lower cost) are ranked consistently, while ambiguous cases are not forced into arbitrary orderings.
>
> Therefore, the benefit of the monotone score mainly comes from two aspects. **First, it provides imagination-time planning with a more stable and fine-grained ranking signal than a fixed penalty**; in our framework, this signal is further introduced into the world model and long-horizon planning through distillation and the ScoreCritic, rather than remaining as an external scorer (Sec. 3.4–3.5). **Second, monotonicity itself provides an important structural prior**: increasing reward should not decrease preference, and increasing cost should not increase preference. This prior makes the learned score less likely to produce rankings that violate the basic semantics of safe RL, even when data are limited and the trade-off is complex. We will make this intuition clearer in the revision.
>
> **(2) What does “structured pair construction” mean?**
> We agree that the term is not introduced clearly enough. In Sec. 3.3, we actually already describe its core mechanism: pairwise comparisons are **not** constructed uniformly at random from global samples, but are formed mainly among **continuations that share the same imagination root**. The purpose is to make comparisons reflect differences in outcomes caused by policy or dynamics variations **from the same starting point**, rather than being overwhelmed by noise from mismatched initial states. The relevant description appears in Sec. 3.3 in the discussion of within-root comparisons and partial-order construction (lines 162–174). In the camera-ready, we will replace this term with a more direct phrase, such as **“within-root dominance/tie pair construction.”**
>
> **(3) On the use of “unsupervised” versus “self-supervised”**
> We accept this comment. **Self-supervised**is the more precise term, since the supervision signal is automatically constructed from reward/cost returns rather than human preference labels. We will revise the wording throughout the paper for consistency.
>
> **(4) Why does including $h(z)$ still preserve monotonicity?**
> Our monotonicity is defined with respect to the explicit inputs $r$ and $c$, while **holding $z$ fixed**. In Eq. (8), $g_r(r)$ depends only on $r$, $g_c(c)$ only on $c$, and $h(z)$ only on $z$. Therefore, $h(z)$ can shift the score baseline based on state context, but it **cannot change the monotone directions**with respect to reward or cost. In other words, $h(z)$ captures state-dependent nuance, not sign changes in $\partial f / \partial r$ or $\partial f / \partial c$. We will make this fixed-$z$ interpretation more explicit. And you can see more monotone parameterization details on Reviewer BPMX Q(1).
>
> **(5) Why is $L_{\text{dom}}$ asymmetric?**
> The asymmetry is intentional, because $L_{\text{dom}}$ is defined on a **directed dominance pair** $(i, j)$. Our paper defines that when $G_i^r \geq G_j^r + \epsilon_r$ and $G_i^c \leq G_j^c - \epsilon_c$, trajectory $i$ dominates $j$, and Eq. (9) enforces $u_i - u_j \geq m$ via a hinge loss. Thus, the directionality comes directly from the dominance relation. If $j$ dominates $i$, the pair order is reversed; if neither dominates the other, the pair is handled by the tie/incomparable objective in Eq. (10). We will clarify this more directly before Eq. (9).
>
> We thank the reviewer again for the detailed comments. We believe that, with these clarifications, both the motivation and the technical details of our method will become much clearer.

---

> > ### Author Rebuttal · Reviewer_2WYJ · 2026-04-03
> >
> > I thank the authors for their detailed rebuttal. It cleared many of my concerns, which were more clarity oriented than soundness or relevance.
> >
> > If the camera ready version indeed includes the clarifications described here, I believe this paper is worthy of acceptance. I am raising my score accordingly.

---

> > > ### Author Response · Authors · 2026-04-05
> > >
> > > Thank you very much for the thoughtful follow-up. We are glad that our clarification helped address your concerns. We sincerely appreciate your constructive feedback, and we will incorporate these clarifications into the camera-ready version.

---

### Decision · Program_Chairs · 2026-04-30

**Decision:**

Accept (regular)

**Comment:**

The manuscript has been received predominantly positively.

The rebuttal, in particular, was convincing.

The paper is considered an important contribution.

However, the authors still need to correct two serious errors:

The references
* Liu, C., Li, Z., Ma, Y., and Darvish, K. Safety-aware trajectory optimization for reinforcement learning. arXiv preprint arXiv:2303.17671, 2023. URL https://arxiv.org/abs/2303.17671.
* Vlastelica, M., Bogdan, M., and Bühler, M. Leveraging privileged information for safe reinforcement learning. In Proceedings of the IEEE International Conference on Robotics and Automation (ICRA), 2023. URL https://arxiv.org/abs/2303.09669.

are incorrect. First, the arXiv links are wrong; second, the references cannot be found. It is therefore necessary to correct these references or to replace them with others.